# OmniCVR: A Benchmark for Omni-Composed Video Retrieval with Vision, Audio, and Text

**Junyang Ji**[1,2,3,*] **Shengjun Zhang**[1], **Da Li**[2,4], **Yuxiao Luo**[2,5], **Yan Wang**[2], **Di Xu**[2], **Biao Yang**[2], **Wei Yuan**[2,†] **Fan Yang**[2,†] **Zhihai He**[3,‡] **Wenming Yang**[1,‡]

[1]Tsinghua University, [2]Kuaishou Technology, [3]Southern University of Science and Technology, [4]University of Chinese Academy of Sciences,    [5]Peking University
hezh@sustech.edu.cn    yang.wenming@sz.tsinghua.edu.cn

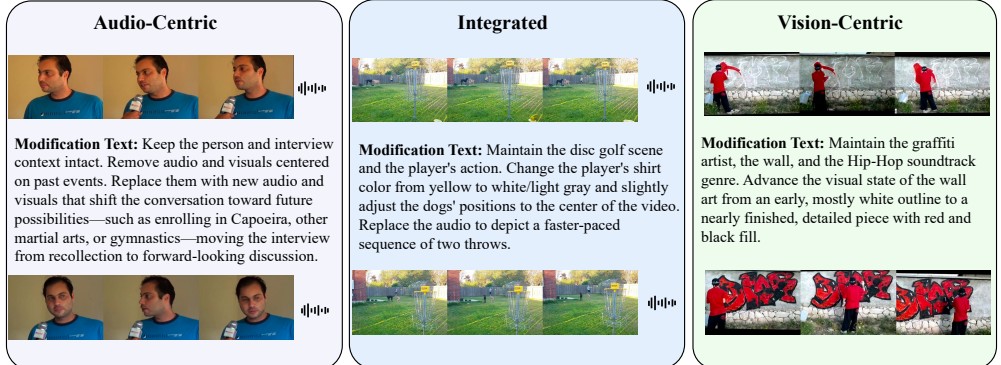

Figure 1: Overview of the OmniCVR Benchmark.

## Abstract

Composed video retrieval presents a complex challenge: retrieving a target video based on a source video and a textual modification instruction. This task demands fine-grained reasoning over multimodal transformations. However, existing benchmarks predominantly focus on vision–text alignment, largely overlooking the rich semantic signals embedded in audio—such as speech, music, and environmental sounds—which are often decisive for comprehensive video understanding. To bridge this gap, we introduce **OmniCVR**, a large-scale benchmark for omni-composed video retrieval that establishes vision, audio, and text as first-class modalities. OmniCVR is constructed via a scalable, automated pipeline integrating content-aware segmentation, omni-modal annotation, and a rigorous dual-validation protocol involving both large language models and human experts. The benchmark comprises vision-centric, audio-centric, and integrated queries, with the latter forming the majority to accurately reflect real-world multimodal complexity. Furthermore, we propose **AudioVLM2Vec**, an audio-aware extension of VLM2Vec. By incorporating explicit audio semantics, AudioVLM2Vec achieves state-of-the-art performance, highlighting fundamental limitations in the audio reasoning capabilities of current multimodal retrieval systems. Our data will be available at https://huggingface.co/datasets/Jun-Yang/OmniCVR.

## 1 Introduction

Video has established itself as the dominant medium for global communication, education, and entertainment, precipitating an exponential growth in digital video content. This deluge has necessitated retrieval systems capable of searching vast repositories with both accuracy and efficiency.

---

*Work done during an internship at Kuaishou Technology.

†Project leaders .

‡Corresponding authors.

While early content-based retrieval depended on low-level visual features, the advent of large-scale vision–language models (Zhan et al., 2024; Kelly et al., 2024; Peng et al., 2024) has revolutionized the field, enabling robust text-to-video retrieval (Radford et al., 2021). Foundational benchmarks such as MSR-VTT (Xu et al., 2016), VATEX (Wang et al., 2019), and YouCook2 (Zhou et al., 2018) have been instrumental in this evolution, pairing extensive video collections with natural language captions to advance video–language alignment.

Recently, the paradigm has shifted toward *composed video retrieval* (CoVR), which requires models to retrieve a target video given a source video and a specific textual modification instruction (Thawakar et al., 2024; Hummel et al., 2024; Thawakar et al., 2025). This formulation demands not only visual grounding but also fine-grained compositional reasoning—for instance, "retrieve the same cooking scene but with a different ingredient." Such benchmarks have successfully pushed the boundary from simple retrieval toward complex reasoning tasks.

Despite these advancements, a critical limitation remains: existing benchmarks overwhelmingly treat video as a purely visual–textual medium, neglecting the audio stream. Audio often carries semantic weight equal to or greater than vision; speech conveys intent, background music establishes mood, and environmental sounds define context. A scene depicting "a crowd cheering at a sports arena" is incompletely represented by visuals alone. By ignoring audio, current benchmarks fail to evaluate models in scenarios where auditory information is decisive. Furthermore, no existing framework systematically addresses retrieval tasks requiring simultaneous modifications across both vision and audio.

To address this deficiency, we present **OmniCVR: the first benchmark for omni-composed video retrieval**, treating vision, audio, and text as unified, first-class modalities. OmniCVR introduces large-scale, compositional retrieval tasks spanning three distinct categories, as illustrated in Figure 1: vision-centric (modifying actions, objects, or scenes), audio-centric (altering music, sound effects, or speech while preserving visual similarity), and integrated (simultaneously modifying both modalities). Unlike prior works, integrated queries dominate OmniCVR, reflecting the intricate multimodal nature of real-world video. The benchmark is constructed via a scalable automated pipeline that combines segmentation, omni-modal annotation, and a dual-validation mechanism (utilizing Gemini 2.5 Pro and human experts in an AND-gated protocol) to ensure both breadth and high-quality data.

In summary, our primary contributions are:

1. We introduce **OmniCVR**, the inaugural large-scale benchmark for omni-modal composed video retrieval, comprising 50K triplets derived from 160K clips and a rigorously validated 5K-instance gold-standard test set.

2. We propose a scalable data generation pipeline integrating content-aware video segmentation, omni-modal annotation, and dual validation, yielding high-quality, compositional instructions.

3. We evaluate seven baselines and propose **AudioVLM2Vec**, which achieves state-of-the-art results on OmniCVR, revealing significant gaps in existing methods regarding audio-centric and compositional reasoning.

## 2 RELATED WORK

### 2.1 VIDEO-TEXT RETRIEVAL BENCHMARKS

The cornerstone of modern video retrieval research is the availability of large-scale video-text datasets. MSR-VTT Xu et al. (2016) pioneered this space with 10,000 web video clips, followed by VATEX Wang et al. (2019), which expanded the scale to over 41,000 clips with bilingual captions. Domain-specific datasets such as YouCook2 Zhou et al. (2018) (instructional cooking) and Charades Sigurdsson et al. (2016) (indoor activities) further diversified the field. Similarly, recent works like MultiVENT 2.0 Kriz et al. (2025) have scaled retrieval to massive multilingual and event-centric domains. While these benchmarks have been instrumental in advancing video-language understanding, they predominantly focus on visual content, largely neglecting the auditory modality. Consequently, they fail to incorporate Composed Video Retrieval scenarios involving audio modifications, such as keeping the visual scene but changing the background music.

## 2.2 COMPOSED VIDEO RETRIEVAL BENCHMARKS

To transcend simple text-based retrieval, the task of Composed Video Retrieval was introduced (Ventura et al., 2024; Gupta et al., 2025; Yue et al., 2025). CoVR tasks a model with retrieving a target video given a source video (or image) and a textual instruction detailing the desired modification. **WebVid-CoVR** Thawakar et al. (2024) established a large-scale, synthetic dataset for this purpose. Subsequently, **Dense-WebVid-CoVR** Thawakar et al. (2025) utilized GPT-4o to generate more elaborate modification texts. In the egocentric domain, **EgoCVR** Hummel et al. (2024) focused on subtle, temporal, and action-oriented modifications. As detailed in Figure 2 and Table 1, a recurrent limitation among these CoVR benchmarks is their exclusive focus on visual modifications, leaving the auditory dimension unexplored. OmniCVR is the first to introduce compositional queries involving audio changes, such as "change the background music to an upbeat pop track."

Table 1: Comparison of OmniCVR with existing video retrieval benchmarks. OmniCVR is the first to explicitly incorporate a detailed, searchable audio modality and support composed audio-visual queries.

| Benchmark | Data Source | Annotation Method | Scale | Task Focus | Audio Modality | CVR |
|---|---|---|---|---|---|---|
| MSR-VTT Xu et al. (2016) | Web Videos | Crowdsourced | 10K clips | Text-to-Video | ✓ | ✗ |
| VATEX Wang et al. (2019) | Web Videos | Crowdsourced | 41K clips | Text-to-Video | ✓ | ✗ |
| WebVid-CoVR Thawakar et al. (2024) | WebVid10M | Synthetic (LLM) | 470K triplets | Visual Comp. | ✗ | ✓ |
| EgoCVR Hummel et al. (2024) | Ego4D | Manual | 2.3K queries | Temporal Comp. | ✗ | ✓ |
| Dense-WebVid-CoVR Thawakar et al. (2025) | WebVid | Synthetic (GPT-4o) | 1.6M samples | Fine-grained Vis. | ✗ | ✓ |
| **OmniCVR (Ours)** | Diverse | Generative (Qwen-Omni) | 160K+ clips | **Omni-Composed** | ✓ | ✓ |

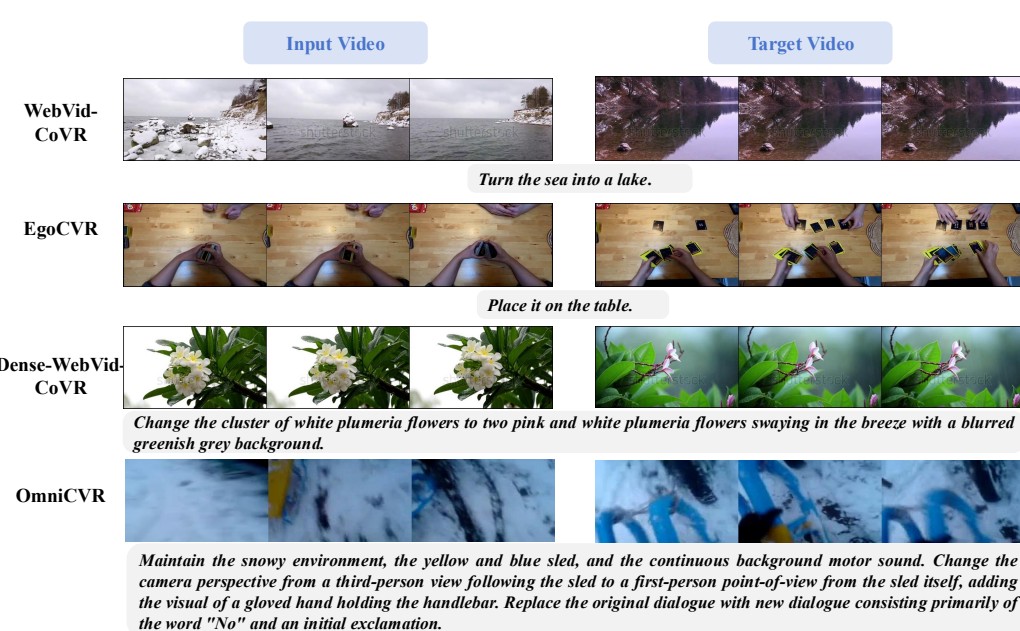

Figure 2: Comparison of OmniCVR with existing video retrieval benchmarks.

## 2.3 AUDIO–VISUAL LEARNING

Beyond video–text retrieval, there is growing interest in joint vision-audio learning. Tasks such as *audio–visual source separation* Afouras et al. (2018) and *audio–visual event localization* Tian et al. (2018) demonstrate the efficacy of multimodal fusion. Benchmarks like **AV-SUPERB** Tseng et al. (2024) evaluate multi-task representations across speech and sound, while recent long-video benchmarks like **MLVU** Zhou et al. (2025) emphasize the challenges of integrating extended temporal and multimodal information. However, these efforts are restricted primarily to classification or grounding; none provide *compositional retrieval tasks* that explicitly require models to adhere to natural language instructions spanning both audio and vision.

## 3 THE OMNICVR BENCHMARK

### 3.1 OVERVIEW OF OMNICVR

We introduce the Omni Composed Video Retrieval Benchmark (OmniCVR), a large-scale omni-modal framework designed to evaluate the compositional retrieval capabilities of multimodal foundation models. Unlike prior CoVR benchmarks focused solely on vision, OmniCVR systematically integrates **video, audio, and text**. Each instance is represented as a triplet: *(source video, modification text, target video)*, where the text specifies the transformation required to map the source to the target.

We curated over **50,000 original long-form videos**, segmenting them into **160,000 coherent short clips**. These clips are annotated and paired to produce **50,000 compositional triplets**. From this corpus, a **5,000-triplet gold-standard test set** was selected and manually validated. OmniCVR tasks fall into three categories:

- **Vision-Centric:** Queries focusing on modifying actions, objects, or scenes.
- **Audio-Centric:** Queries altering music, sound effects, or speech while maintaining high visual similarity.
- **Integrated:** Queries requiring simultaneous modifications across both visual and auditory modalities.

This design ensures OmniCVR evaluates not only perceptual grounding but also the capacity to follow complex multimodal modification instructions.

Table 2: Core statistics of OmniCVR. Integrated queries dominate the benchmark, reflecting the focus on realistic cross-modal modifications.

| Statistics | Number |
|---|---|
| Training Triplets | 45k+ |
| Unique Video Clips | 160k+ |
| Test Set (gold-standard) | 5,000 |
| Query Types (Vision:Audio:Integrated) | 22.82% : 20.00% : 57.18% |
| Avg. Query Length | 52.6 words |
| Vocabulary Size | 25k+ |
| Avg. Video Length | 11.8 sec |

Table 2 summarizes the benchmark statistics. Unlike existing datasets, OmniCVR emphasizes **integrated queries**, which constitute the majority of tasks. This distribution mirrors real-world complexity, where modifications rarely occur in isolation. Vision-centric queries form the second largest cohort, while audio-centric queries are rarer due to strict pairing constraints (high visual similarity, low audio similarity). It is worth noting that the average query length in OmniCVR (52.6 words) is higher than in visual-only CVR benchmarks. This is a deliberate design choice: Integrated queries (57.18% of the dataset) require specifying simultaneous transformations in both visual and auditory domains to avoid ambiguity. Unlike prior works that overlook audio, OmniCVR necessitates denser descriptions to capture the full spectrum of multimodal evolution, reflecting the complexity of real-world video editing and retrieval scenarios.

To further demonstrate the semantic breadth of the benchmark, Figure 3 visualizes the hierarchical distribution of content across modalities. Regarding video content (Figure 3a), OmniCVR encompasses four primary domains: *Entertainment & Events*, *Instructional & Procedural*, *Daily Life, Nature & Travel*, and *Music & Performance*. These are structured into granular subcategories—ranging from *Culinary Arts* to *Wilderness & Wildlife*—to ensure robust coverage of diverse visual scenes. Similarly, the auditory landscape is rigorously balanced, as shown in Figure 3b. We organize audio instances into a two-level taxonomy comprising three high-level classes (*Speech*, *Music*, and *Sound*) branched into 15 distinct subcategories (e.g., *Scripted Dialogue*, *Instrumental Music*, and *Nature & Animal Sounds*). This fine-grained categorization highlights the acoustic richness and real-world complexity embedded in our benchmark.

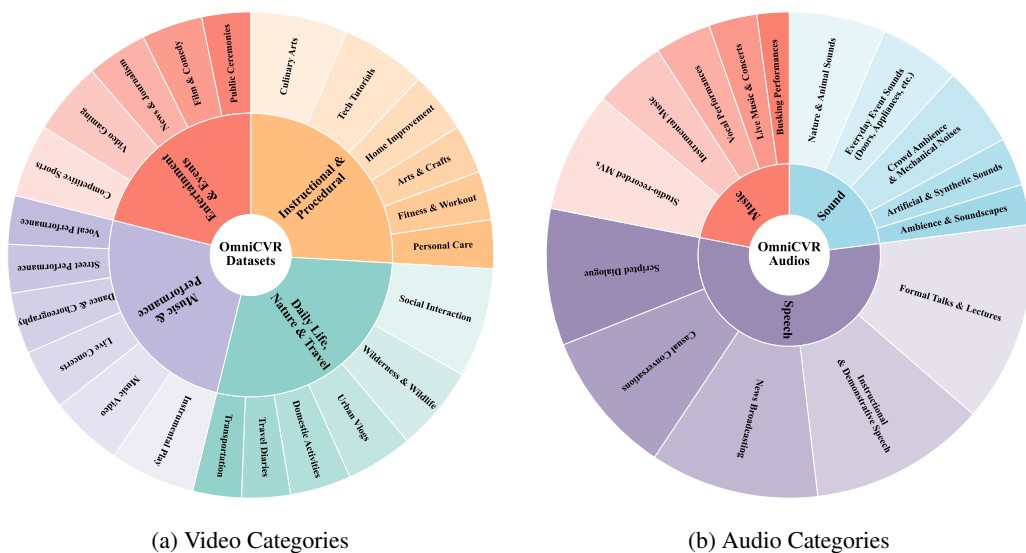

(a) Video Categories    (b) Audio Categories

Figure 3: **The data diversity of OmniCVR.** (a) Illustrates the hierarchical taxonomy of video content, spanning four major domains and their subcategories. (b) Displays the distribution of audio types, categorized into Speech, Music, and Sound with fine-grained subdivisions.

## 3.2 DATA GENERATION PIPELINE

OmniCVR is constructed via a three-stage automated pipeline, depicted in Figure 4, ensuring both scale and high fidelity.

**Stage 1: Video Curation and Segmentation.** We begin by collecting long-form videos (typically ranging from several minutes to multiple hours) from a diverse set of public datasets, including HowTo100M (Miech et al., 2019), MSR-VTT, VATEX, and others. These sources cover diverse domains such as instructional content, daily activities, and entertainment.

From these long-form videos, we extract semantically coherent short clips (5–15 seconds, average 11.8 seconds) that serve as the final retrieval units. We employ `PySceneDetect` to define segments based on inter-frame HSV differences ($\tau = 36$), effectively capturing scene changes while ignoring minor camera motion. Post-segmentation, we filter clips using two metrics: **Action Intensity** (optical flow magnitude) and **Scene Richness** (visual feature variance). Only clips exceeding a combined threshold are retained, ensuring semantic density.

**Stage 2: Generative Omni-Modal Annotation.** Clips are annotated using `Qwen2.5-Omni`, which jointly encodes video and audio. To enhance auditory detail, we integrate automatic **audio transcription**. The annotation prompt requests structured descriptions of scenes, actions, objects, and audio events. Specifically for audio, to capture the full acoustic spectrum, we enforce a strict schema covering **para-linguistic features, lexical content, environmental hierarchy, and temporal dynamics** (see Appendix G for full details). These dimensions are validated in Stage 3 to ensure retrieval is grounded in fine-grained audio semantics. Quality is enforced through a two-stage verification process: automatic consistency checks with `Gemini 2.5 Pro` and manual expert review.

**Stage 3: Triplet Mining for Compositional Retrieval.** We generate triplets via three strategies:

- **Vision-Centric:** Constructed either from (a) different segments of the same long video, or (b) distinct clips from the same video source, ensuring coverage of both coarse-grained and fine-grained visual differences. Audio is preserved to isolate visual reasoning.

- **Audio-Centric:** Candidate pairs are first filtered by requiring high visual similarity (video CLIP cosine similarity $> 0.9$). Among these, pairs with low auditory similarity (audio

embedding cosine similarity $< 0.3$, measured by the CLAP model) are selected. This guarantees that the visual scene remains constant while audio varies significantly.

- **Integrated:** Pairs are chosen to differ in both modalities, with low similarity in CLIP embeddings (vision) and low similarity in CLAP embeddings (audio).

Modification texts are generated by prompting an LLM with the structured annotations of the source and target, explicitly encoding the relevant differences.

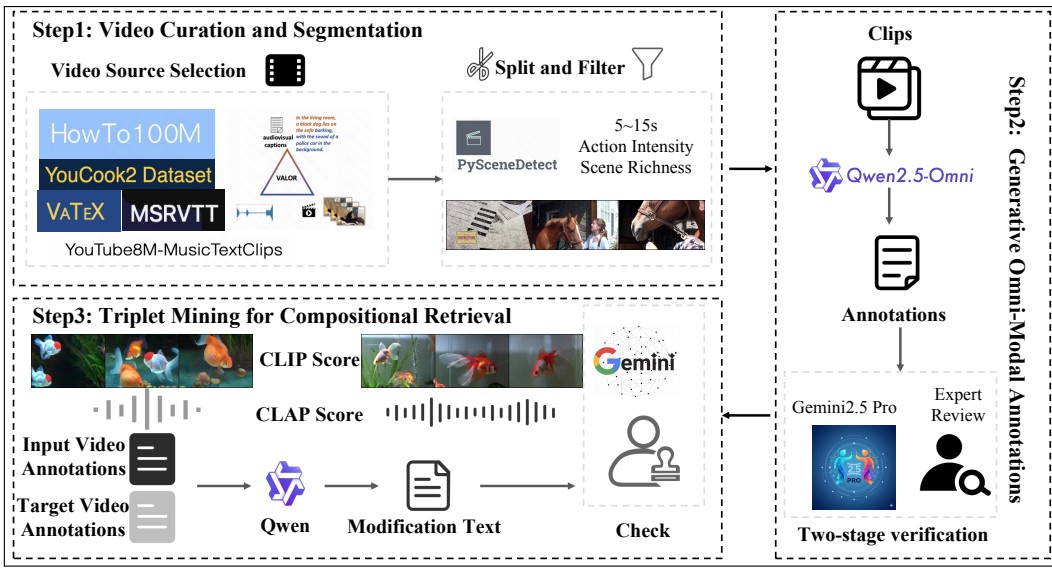

Figure 4: OmniCVR Benchmark Construction Pipeline.

## 3.3 EVALUATION AND TEST SET CURATION

We construct a gold-standard test set of **5,000** instances using a *concurrent dual-gate protocol*. For *every* candidate triplet, both **Gemini 2.5 Pro** and a **human expert** independently validate the paired videos and modification text. A triplet is admitted only if *both* approve. This AND-gated review guarantees semantic fidelity and consistency. The resulting test set preserves the natural query distribution (Integrated > Vision > Audio), creating a realistic evaluation regime.

Table 3: Distribution of audio categories (speech, music, sound) in the source and target videos.

| Video Type | Speech (%) | Music (%) | Sound (%) |
|---|---|---|---|
| Source video | 56 | 23 | 21 |
| Target video | 55 | 23 | 22 |

Table 3 illustrates the modality distribution of the audio streams in the source and target videos. We decompose each video's soundtrack into three categories: *speech*, *music*, and *sound*.

## 4 EXPERIMENTAL SETUP

### 4.1 TASKS AND EVALUATION

OmniCVR is designed to rigorously assess compositional retrieval across multiple modalities, focusing on **Composed Video Retrieval (CVR)**. In this task, a model is given a source video and a natural language instruction and must retrieve the corresponding target video from a candidate pool. Unlike traditional video retrieval, which emphasizes semantic similarity, CVR tests a model's ability to reason over transformations like object changes or action alterations. Queries are categorized

into three groups: vision-only modifications (focused on visual appearance and motion), audio-only modifications (based on acoustic cues like speech or background music), and joint vision–audio modifications (requiring integrated multimodal reasoning). Performance is measured using standard retrieval metrics—Recall at K (R@1, R@5, R@10)—which reflect the accuracy of retrieving the correct target from the top candidates. This provides a challenging evaluation that mirrors real-world multimodal retrieval tasks.

## 4.2 BASELINE MODELS

**Large Multimodal Embedding Models.** (i) **OmniEmbed-v0.1-multivent** (Ma et al., 2025): A unified model from Tevatron 2.0 that encodes text, image, audio, and video into a shared space, achieving state-of-the-art performance in cross-modal video retrieval, particularly on the MAGMaR Shared Task (Zhan et al., 2025), after fine-tuning on MultiVENT data (Kriz et al., 2025) with joint vision-audio-text supervision. (ii) **VLM2Vec** (Jiang et al., 2025): A framework that converts large vision-language models (e.g., Qwen2-VL (Wang et al., 2024), LLaVA (Liu et al., 2023), Phi-3.5-V) into universal embedding models through contrastive learning on the MMEB benchmark. VLM2Vec outperforms baselines like CLIP (Radford et al., 2021) and BLIP (Li et al., 2022), showing 10-20% improvement in retrieval tasks. (iii) **AudioVLM2Vec (Ours)**: Our extension of VLM2Vec that integrates audio semantics by using **Qwen2-Audio** (Chu et al., 2024) to generate fine-grained captions of video audio tracks. The captions are combined with user queries and fed into VLM2Vec, enhancing its representation power while focusing on audio signals.

**Lightweight and Task-Specific Models.** (iv) **CLIP** (Radford et al., 2021): A foundational vision-language model trained on large image-text pairs, adapted for video retrieval via frame-level temporal averaging. (v) **CoVR** (Thawakar et al., 2024): A video retrieval model optimized for vision-centric modifications, evaluated for its generalization beyond visual changes. (vi) **BLIP** (Li et al., 2022): A unified vision-language model adapted for retrieval tasks using its vision-language matching head. (vii) **ImageBind** (Girdhar et al., 2023): A model learning a joint embedding across six modalities, including audio and video, ideal for audio-centric tasks.

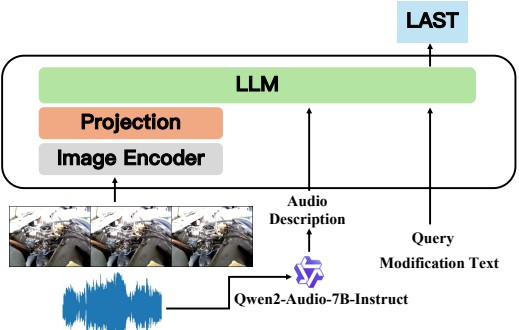

Figure 5: The framework of AudioVLM2Vec.

Figure 5 illustrates the design of our proposed **AudioVLM2Vec** model. The framework extends VLM2Vec by explicitly injecting audio semantics into the retrieval pipeline. Given a video, we first encode its visual content using a pretrained image encoder followed by a lightweight projection layer. In parallel, the audio track is processed by **Qwen2-Audio-7B-Instruct**, which generates a fine-grained natural language description of the acoustic scene. This audio-derived text is concatenated with the user's modification query and then fed into the LLM backbone, ensuring that both vision and audio cues are aligned in a shared semantic space. By feeding the transcribed audio semantics alongside visual tokens into the LLM, we leverage the model's multi-head self-attention mechanism to jointly process both modalities. This allows the model to learn synergistic and causal relationships (e.g., aligning the text 'lips moving' with corresponding visual tokens) within a shared high-dimensional semantic space. The final multimodal embedding is obtained from the LLM and optimized for retrieval via contrastive learning. By translating audio signals into text and integrating them at the embedding stage, AudioVLM2Vec effectively grounds compositional queries in both modalities, yielding substantial gains on audio-centric and cross-modal retrieval tasks.

## 4.3 EVALUATION STRATEGY

For each query, we compute similarity scores between the query embedding and candidate video embeddings, ranking candidates accordingly. To mitigate potential variance introduced by candidate set composition, we shuffle candidate pools five times and report averaged metrics. For audio-centric tasks, we additionally control for modality imbalance by ensuring candidate pools always include visually similar but acoustically distinct distractors, and vice versa. This evaluation protocol ensures fairness across different model families while highlighting the challenges of multimodal compositional retrieval.

## 5 RESULTS AND DISCUSSION

### 5.1 MAIN RESULTS

Table 4 presents Recall@K on OmniCVR across overall queries, while Table 5 isolates the case of *audio-centric queries*. We summarize the main findings below. The symmetric vision-centric retrieval results on OmniCVR are reported in Table 10 in Appendix C.

1. **Audio-centric queries expose a critical failure mode in existing baselines.** While our AudioVLM2Vec adapts robustly to audio-dependent queries (77.2 R@1), strong baselines suffer catastrophic performance degradation. For instance, VLM2Vec drops from an overall R@1 of 38.44 to just 12.4 in the audio-centric setting. This disparity underscores the unique difficulty of audio compositionality compared to visual tasks, a challenge that prior models fail to address.

2. **Large-scale multimodal models establish a superior performance tier.** Across all query types, large embedding models consistently and substantially outperform lightweight, task-specific retrievers, validating the efficacy of large-scale pre-training for compositional reasoning.

3. **AudioVLM2Vec achieves universal state-of-the-art performance.** Our model ranks first across all categories and $K$ values, achieving **66.98** overall R@1 and **77.2** on audio-centric queries. This consistency confirms that our architecture generalizes effectively across both unimodal and integrated multimodal retrieval tasks.

4. **Explicit audio semantics are the decisive factor for performance gains.** The impact of injecting audio descriptions is profound: AudioVLM2Vec surpasses the VLM2Vec baseline by an impressive **+64.8** absolute points (77.2 vs. 12.4) in the audio-centric setting and **+28.5** points overall. These results demonstrate that audio-aware embeddings are not merely beneficial but indispensable for handling compositional queries involving non-visual transformations.

Table 4: Overall performance comparison of baseline models on OmniCVR. We report Recall at K (R@1, R@3, R@5, R@10). Best results within each group are highlighted in **bold**, and second-best are underlined.

| Models | Backbone | R@1 | R@3 | R@5 | R@10 |
|---|---|---|---|---|---|
| **Lightweight and Task-Specific Models** | | | | | |
| CLIP | CLIP | 27.54 | 50.46 | 56.70 | 62.62 |
| CoVR | BLIP2 (Li et al., 2023) | 11.46 | 22.88 | 28.08 | 35.18 |
| BLIP | BLIP | 6.3 | 11.84 | 14.12 | 17.00 |
| IMAGEBIND | CLIP | 17.28 | 29.55 | 43.34 | 45.33 |
| **Large Multimodal Embedding Models** | | | | | |
| OmniEmbed-v0.1-multivent | Qwen2.5-Omni | 31.90 | 51.50 | 57.04 | 64.00 |
| VLM2Vec | Qwen2-VL | 38.44 | 55.48 | 60.44 | 66.60 |
| AudioVLM2Vec (Ours) | Qwen2-Audio + Qwen2-VL | **66.98** | **77.84** | **80.86** | **84.40** |

Table 5: Audio-centric retrieval performance of Large Multimodal Embedding Models on OmniCVR. We report Recall at K (R@1, R@3, R@5, R@10). Best results are highlighted in **bold**.

| Models | Backbone | R@1 | R@3 | R@5 | R@10 |
|---|---|---|---|---|---|
| OmniEmbed-v0.1-multivent | Qwen2.5-Omni | 13.6 | 28.5 | 35.8 | 47.0 |
| VLM2Vec | Qwen2-VL | 12.4 | 23.3 | 30.4 | 42.3 |
| AudioVLM2Vec (Ours) | Qwen2-Audio + Qwen2-VL | **77.2** | **87.3** | **90.7** | **94.2** |

## 5.2 DETAILED ANALYSIS AND ABLATION STUDIES

**Breakdown of Audio-Centric Performance.** To better understand the model's capabilities across different acoustic domains, we decompose the audio-centric performance by target audio category (Human Speech, Music, and Sound). As shown in Table 6, AudioVLM2Vec achieves dominant performance in Human Speech (+85.23% gain) and Music (+70.36% gain). This indicates that converting audio to text effectively captures both lexical content and para-linguistic features like genre and mood. The gain is slightly lower but still substantial for Sounds (+49.56%), reflecting the inherent challenge in describing unstructured acoustic events compared to structured speech or music.

Table 6: Fine-grained breakdown of audio-centric retrieval (R@1) on OmniCVR by target audio category. Best results are highlighted in **bold**.

| Target Audio Category | VLM2Vec | AudioVLM2Vec (Ours) | Absolute Gain |
|---|---|---|---|
| Human Speech | 11.36 | **96.59** | **+85.23** |
| Music | 16.07 | **86.43** | **+70.36** |
| Sound | 10.75 | **60.31** | **+49.56** |

**Impact of Source Video.** To determine if detailed modification texts render the source video redundant (effectively reducing the task to text-to-video retrieval), we conducted a "Blind Retrieval" ablation. Here, we removed the visual frames of the source video, forcing the model to rely solely on the modification instruction and source audio. As shown in Table 7, performance drops catastrophically without the source video. For AudioVLM2Vec on audio-centric queries, R@1 plummets by 49.1% (from 77.20% to 28.10%). This confirms that the modification text functions as a relative instruction rather than a standalone description. The source video provides essential context (e.g., the visual scene "park") to filter distractors, validating that OmniCVR rigorously evaluates compositional reasoning.

Table 7: Ablation study on the importance of source video for VLM2Vec on OmniCVR audio-centric retrieval.

| Metric | VLM2Vec (With Source Video) | Blind / Text-Only (No Source Video) | Performance Drop ($\Delta$) |
|---|---|---|---|
| R@1 | **77.20%** | 28.10% | -49.10% |
| R@3 | **87.30%** | 33.20% | -54.10% |
| R@5 | **90.70%** | 42.50% | -48.20% |
| R@10 | **94.20%** | 57.80% | -36.40% |

**Efficiency Analysis.** We benchmarked inference latency under the same hardware configuration as used in VLM2Vec (Jiang et al., 2025), evaluating performance on 10-second video inputs. While AudioVLM2Vec increases latency from 1.72s (VLM2Vec) to 4.77s (approx. 1.77x increase in processing overhead due to audio transcription), this trade-off yields a 64.8% improvement in audio-centric retrieval accuracy. Furthermore, with a Real-Time Factor (RTF) of $\tilde{0}.5$, the system remains faster than real-time playback, ensuring deployability.

**Native Audio Tokens vs. Audio-as-Text.** To rigorously isolate the contribution of our Audio-as-Text fusion mechanism from backbone differences, we performed a controlled ablation on **OmniEmbed-v0.1-multivent**. This model's backbone (**Qwen2.5-Omni**) natively accepts raw audio

waveforms via a dedicated audio tower. We compared the original model (audio tower enabled) against a modified version where the audio tower is disabled and replaced by our Qwen2-Audio-7B-Instruct transcribed captions.

As shown in Table 8, simply replacing latent audio tokens with explicit textual descriptions—while keeping the backbone, projector, and all training data identical—yields a dramatic improvement from 13.6 to **32.7** on R@1 (+19.1 absolute points, 2.4× relative gain).

Table 8: Controlled ablation on OmniEmbed-v0.1-multivent backbone: native audio tokens vs. our Audio-as-Text fusion mechanism on audio-centric retrieval. Best results are highlighted in **bold**.

| Model Setting | Audio Mechanism | R@1 | R@3 | R@5 | R@10 |
|---|---|---|---|---|---|
| OmniEmbed (Original) | Native Audio Tokens | 13.6 | 28.5 | 35.8 | 47.0 |
| OmniEmbed (Modified) | Audio-as-Text (Ours) | **32.7** | **48.0** | **58.9** | **69.1** |

We further break down the results of this ablation by target audio category. As shown in Table 9, our Audio-as-Text strategy delivers **consistent and substantial improvements across all categories**, including a near-doubling of R@1 on Music (+13.93 points) and Sound (+13.37 points). These results demonstrates that explicitly converting audio into dense, semantically rich captions—rather than relying on latent audio tokens—provides a far more effective and universal audio representation, successfully capturing non-lexical attributes such as musical genre, timbre, rhythm, and complex environmental events.

Table 9: Per-category breakdown of the controlled OmniEmbed ablation (R@1). Replacing native audio tokens with our Audio-as-Text mechanism yields large gains **across all acoustic domains**, including non-speech categories. Best results are highlighted in **bold**.

| Target Audio Category | OmniEmbed (Original) | OmniEmbed (Ours-modified) | Absolute Gain |
|---|---|---|---|
| Human Speech | 11.36 | **50.38** | +39.02 |
| Music | 16.07 | **30.00** | +13.93 |
| Sound | 10.75 | **24.12** | +13.37 |

## 6   CONCLUSION, LIMITATIONS AND FUTURE WORK

In this paper, we introduced **OmniCVR**, a large-scale benchmark for *omni-composed video retrieval* that establishes vision, audio, and text as first-class modalities. By requiring models to retrieve target videos based on source videos and natural-language instructions—spanning vision-centric, audio-centric, and integrated queries—OmniCVR provides a rigorous testbed for multimodal compositionality. Our systematic evaluation reveals that existing state-of-the-art retrievers significantly underutilize acoustic information, struggling when retrieval hinges on non-visual transformations. To address this, we proposed **AudioVLM2Vec**, which injects explicit audio semantics into the embedding pipeline. This approach achieves state-of-the-art results and exposes the failure of current "full-modality" systems to effectively attend to speech and environmental sound cues.

**Limitations.** Despite its strong performance, a primary limitation is the **inference latency** caused by the intermediate audio-to-text transcription. While effective for semantic bridging, this step incurs higher computational overhead than latent embedding methods. Future work will focus on optimizing this pipeline—potentially via lightweight adaptors or distillation—to accelerate embedding generation for real-time applications.

**Future Work.** Moving forward, we aim to: (i) incorporate additional modalities (e.g., depth, 3D) for richer reasoning; (ii) develop diagnostic protocols to probe temporal grounding and cross-modal consistency; (iii) scale to cinematic long-form videos (ranging from minutes to hours, e.g., full movies/TV episodes) by utilizing dense cropping from complex long contexts to yield short clips with richer temporal and semantic density, thereby introducing harder distractors and larger retrieval pools; (iv) explore open-ended retrieval to study hallucinations; and (v) leverage OmniCVR as a training resource to develop robust end-to-end omni embeddings.

## ACKNOWLEDGMENTS

We sincerely thank the reviewers for their insightful comments and constructive feedback, which have greatly contributed to improving this work. This work was supported by the Special Foundations for the Development of Strategic Emerging Industries of Shenzhen (No. KJZD20231023094700001) and the National Natural Science Foundation of China (No. 62331014). We acknowledge the computational support of the Center for Computational Science and Engineering at Southern University of Science and Technology.

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

## A  DATASETS USED IN STAGE 1 (VIDEO CURATION & SEGMENTATION)

In Stage 1, we curate and segment long-form videos (typically minutes to hours in duration) from six complementary datasets that cover diverse domains, modalities, and annotation styles. From each of these long-form videos, we extract multiple semantically coherent short clips (5–15 seconds) that constitute the final retrieval items in our benchmark (average length 11.8 seconds). Below we provide more detailed descriptions of the six datasets: HowTo100M (Miech et al., 2019), MSR-VTT (Xu et al., 2016), VATEX (Wang et al., 2019), YouTube8m-MusicTextClips (Abu-El-Haija et al., 2016), YouCook2 (Zhou et al., 2018), and VALOR (Liu et al., 2024).

**HowTo100M (Miech et al., 2019).**  A large-scale collection of narrated instructional videos from YouTube. It contains about 1.2M long videos and roughly 136M automatically transcribed narration clips, covering over 23k diverse tasks (cooking, DIY, fitness, crafts, etc.). Strengths include unmatched scale and domain coverage, making it ideal for weakly supervised pretraining of text–video embeddings. However, ASR transcripts are noisy and loosely aligned with visual content, and activity distribution is highly imbalanced. In Stage 1, it serves as the primary source of long videos, where narration timestamps provide coarse cues for segmentation.

**MSR-VTT (Xu et al., 2016).**  A widely used benchmark of general-domain short clips with human-written captions. It contains 10k clips and 200k captions (about 20 per clip). Strengths are high-quality human annotations and balanced coverage across diverse scenarios. Limitations include its clip-level scope and lack of fine-grained temporal structure. In Stage 1, it is primarily used as a reference benchmark to evaluate the semantic quality of curated segments after segmentation.

**VATEX (Wang et al., 2019).**  A multilingual video–caption dataset with both English and Chinese annotations. It consists of 41k videos paired with 825k captions, including 206k English–Chinese parallel sentences. The dataset enables multilingual retrieval and cross-lingual transfer. Its limitations lie in its clip-level nature and absence of dense temporal supervision. In Stage 1, it complements other datasets by providing multilingual benchmarks to test cross-lingual robustness of curated segments.

**YouTube8m-MusicTextClips (Abu-El-Haija et al., 2016).**  Derived from the large-scale YouTube-8M dataset, this subset focuses on music and music-video content aligned with text or tags. YouTube-8M contains millions of videos and billions of frames with noisy machine-generated labels across 4.8k entity categories. Strengths are its massive scale and coverage, especially in entertainment and music. Weaknesses are label noise and lack of dense natural-language captions. In Stage 1, it acts as a supplementary pool and weakly labeled filtering resource for music/entertainment domains.

**YouCook2 (Zhou et al., 2018).**  A domain-specific dataset of cooking videos with step-level annotations. It contains 2k untrimmed YouTube videos (about 176 hours, across 89 recipes), each segmented into procedure steps with textual descriptions. It is a gold-standard benchmark for procedure segmentation and dense video–text alignment. Limitations include its domain restriction to cooking and moderate video length. In Stage 1, it is used as a calibration set for validating segmentation quality and procedure-aware alignment.

**VALOR (Liu et al., 2024).** A tri-modal dataset (vision–audio–language) designed for audiovisual captioning and retrieval. VALOR-1M provides about 1M audiovisual clips for pretraining, and VALOR-32k offers a smaller high-quality evaluation benchmark with human-curated audiovisual captions. Strengths include explicit modeling of audio cues in addition to visual and textual context, enabling tri-modal learning. Limitations include smaller scale compared to HowTo100M and potential annotation subjectivity. In Stage 1, VALOR supports audio-informed segmentation (e.g., boundary detection from speech or sound transitions) and tri-modal retrieval evaluation.

## B    BASELINE MODELS

To contextualize the performance of our approach, we benchmark against a diverse set of baselines, spanning both large multimodal embedding models and lightweight or task-specific retrievers. Below we provide detailed descriptions of each baseline model.

**CLIP (Radford et al., 2021).** A foundational vision–language model trained on large-scale image–text pairs. For video retrieval tasks, we follow standard practice by uniformly sampling 15 frames per video and averaging their frame-level embeddings to obtain the video representation. While CLIP provides a strong baseline for vision–text alignment, it lacks explicit modeling of audio or video-specific temporal dynamics.

**CoVR (Thawakar et al., 2024).** A model specifically designed for composed video retrieval, focusing on scenarios where the query involves modifications of existing video content. CoVR is optimized for vision-centric transformations, and in our setting, we uniformly sample 15 frames per video and average their embeddings to form the video representation. We assess CoVR's ability to generalize beyond purely visual changes. Its lightweight design makes it efficient, though its limited multimodal scope is a constraint for audio-aware retrieval.

**BLIP (Li et al., 2022).** A unified vision–language understanding and generation model. We adapt BLIP for retrieval by using its vision–language matching head to score candidate videos. Following our experimental protocol, we uniformly sample 15 frames per video and average their embeddings to construct the video representation. BLIP demonstrates strong cross-modal reasoning and captioning ability, making it a competitive retrieval baseline. However, like CLIP, it does not natively incorporate audio cues.

**ImageBind (Girdhar et al., 2023).** A multimodal embedding model that learns a joint representation space across six modalities: images, text, audio, video, depth, and IMU signals. For video retrieval, we apply the same protocol as other lightweight baselines, uniformly sampling 15 frames per video and averaging their frame-level embeddings. Its broad modality coverage makes it a natural fit for audio–video retrieval tasks. The ability to align heterogeneous modalities directly in a shared space provides a strong baseline for multimodal integration, though its representations may be less specialized than task-specific models.

**OmniEmbed-v0.1-multivent (Ma et al., 2025).** A unified multimodal embedding model built on the Tevatron 2.0 framework. It is trained to encode text, image, audio, and video into a shared representation space. OmniEmbed-v0.1-multivent has achieved state-of-the-art performance in cross-modal video retrieval benchmarks, such as the MAGMaR Shared Task (Zhan et al., 2025), by fine-tuning on the MultiVENT dataset (Kriz et al., 2025) with joint vision–audio–text supervision. Its strength lies in versatility across modalities, making it a strong baseline for multimodal retrieval.

**VLM2Vec (Jiang et al., 2025).** A general framework that transforms existing large vision–language models (VLMs)—including Qwen2-VL (Wang et al., 2024), LLaVA (Liu et al., 2023), and Phi-3.5-V—into universal embedding models through contrastive learning on the Massive Multimodal Embedding Benchmark (MMEB). VLM2Vec consistently surpasses conventional baselines such as CLIP (Radford et al., 2021) and BLIP (Li et al., 2022), with improvements on retrieval tasks. This approach highlights the effectiveness of adapting powerful pretrained VLMs into embedding-focused architectures.

**AudioVLM2Vec (Ours).** Our proposed extension of VLM2Vec that explicitly integrates audio semantics. We employ Qwen2-Audio (Chu et al., 2024) to generate fine-grained captions from the video's audio track, which are then concatenated with the user's modification query and fed into VLM2Vec. This design leverages the strong representation capabilities of large VLMs while ensuring sensitivity to audio signals, thereby improving performance on audio-centric retrieval scenarios.

## C   SUPPLEMENTARY EXPERIMENTS AND DISCUSSION

### C.1   OMNICVR DATA STATISTICS AND ANALYSIS

Figure 6 shows the distribution of video lengths in the OmniCVR training dataset, where most clips fall between 5 and 15 seconds.

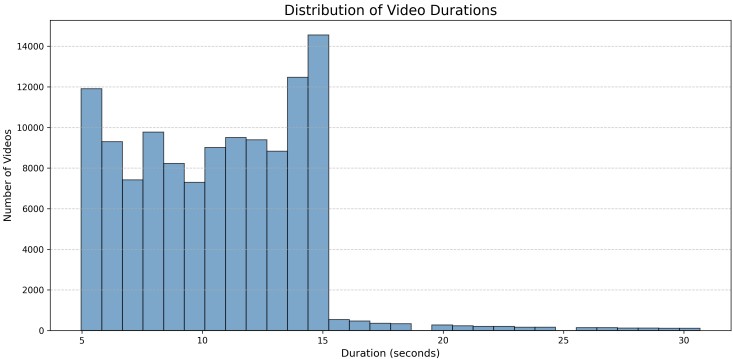

Figure 6: Video length distribution of the OmniCVR training dataset.

To better illustrate the breadth of real-world scenarios covered by OmniCVR, we provide a detailed breakdown of its content sources. OmniCVR aggregates videos from diverse publicly available datasets spanning multiple genres:

- **Instructional & Procedural Content**: Videos depicting step-by-step tasks such as cooking recipes, DIY repairs, crafting tutorials, and fitness routines. These are critical for evaluating a model's ability to understand fine-grained actions and temporal dependencies.

- **Daily Life, Nature & Travel**: Unscripted, in-the-wild footage capturing everyday human activities, natural environments, and travel vlogs. This category provides rich visual and auditory context and serves as the primary source for environmental sound-related queries.

- **Music & Performance**: Clips featuring musical instruments, dance performances, and live concerts. These are specifically leveraged for "Audio-Centric" queries, enabling evaluation of a model's capacity to distinguish between musical genres, tempos, instruments, and performance styles.

- **General Entertainment & Events**: A broad collection of web-sourced clips including sports highlights, news broadcasts, public ceremonies, and other event-driven content, ensuring coverage of dynamic and socially relevant scenarios.

This multi-genre composition ensures that OmniCVR comprehensively reflects the heterogeneity of real-world audiovisual experiences, making it a robust benchmark for cross-modal retrieval and understanding.

Complementing these visual genres, OmniCVR explicitly models the acoustic dimension with high granularity. To ensure comprehensive coverage of acoustic, semantic, and para-linguistic features, we categorize audio content into three primary domains, each enforcing a strict schema of fine-grained attributes:

- **Human Speech**: This category addresses both the *lexical* and *para-linguistic* dimensions of spoken audio. Our annotation pipeline explicitly captures:

- *Lexical Content*: Verbatim transcripts of the speech to ground accurate semantic understanding.
- *Para-linguistic Features*: Identification of speaker characteristics and emotional tone (e.g., neutral, angry, fearful, surprised), distinguishing retrieval targets based on *how* something is said, not just *what* is said.

- **Music**: This category focuses on *temporal dynamics* and stylistic attributes. The annotations provide detailed descriptions of:
  - *Genre & Instrumentation*: Identification of specific musical styles and the instruments present.
  - *Temporal Dynamics*: Chronological descriptions of rhythm, pace, and intensity (e.g., distinguishing a "slow, steady beat" from a "fast, erratic tempo") and the overall atmospheric mood.

- **Environmental Sound**: To address the complexity of acoustic environments, we enforce a detailed *hierarchy* distinguishing between:
  - *Nature Sounds*: Elements such as wind, rain, water flow, and animal calls.
  - *Mechanical & Urban Soundscapes*: Sounds of engines, machinery, alarms, traffic patterns, and construction noise.
  - *Foley & Action Sounds*: Distinct, event-driven sounds such as footsteps, glass breaking, or doors closing.

By explicitly modeling these dimensions during the annotation and verification stages, OmniCVR ensures that retrieval queries are grounded in rich, fine-grained audio semantics rather than simplified labels.

## C.2 VISION-CENTRIC RETRIEVAL RESULTS

Table 10 reports vision-centric retrieval results on OmniCVR. Within lightweight baselines, IMAGE-BIND achieves the best performance, indicating that simple cross-modal alignment still provides competitive vision retrieval ability. Yet, large multimodal embedding models clearly dominate: VLM2Vec already surpasses OmniEmbed-v0.1-multivent, and our AudioVLM2Vec further advances state-of-the-art performance by a large margin (e.g., +14.3% R@1 over VLM2Vec). Interestingly, the consistent gains obtained by AudioVLM2Vec—even under vision-only evaluation—suggest that audio representations implicitly encode complementary visual characteristics, which can be leveraged during joint embedding learning. This finding highlights the underexplored synergy between audio and vision modalities: audio not only enriches multimodal fusion but also strengthens pure visual retrieval through cross-modal feature transfer.

Table 10: Vision-centric retrieval performance on OmniCVR. We report Recall at K (R@1, R@3, R@5, R@10). Best results within each category are highlighted in **bold**.

| Models | Backbone | R@1 | R@3 | R@5 | R@10 |
|---|---|---|---|---|---|
| **Lightweight and Task-Specific Models** | | | | | |
| CLIP | CLIP | 39.53 | 51.18 | 56.27 | 63.10 |
| CoVR | BLIP2 | 2.10 | 6.57 | 9.29 | 14.46 |
| BLIP | BLIP | 6.13 | 8.85 | 10.60 | 13.41 |
| IMAGEBIND | CLIP | **47.85** | **57.58** | **61.09** | **65.82** |
| **Large Multimodal Embedding Models** | | | | | |
| OmniEmbed-v0.1-multivent | Qwen2.5-Omni | 50.74 | 63.45 | 69.85 | 77.21 |
| VLM2Vec | Qwen2-VL | 55.04 | 66.78 | 70.99 | 75.99 |
| AudioVLM2Vec (Ours) | Qwen2-Audio + Qwen2-VL | **69.33** | **76.51** | **80.28** | **82.82** |

## C.3 GENERALIZATION TO OUT-OF-DOMAIN TASKS

To further validate the generalization capability of the proposed audio-augmented representations, we evaluate AudioVLM2Vec in a zero-shot manner on the widely used MSR-VTT dataset (Xu et al.,

2016) for conventional text-to-video retrieval. As shown in Table 11, AudioVLM2Vec consistently outperforms the strong VLM2Vec baseline across all metrics (+1.7–2.3 absolute points), even though MSR-VTT captions rarely mention sound events explicitly. Qualitative analysis reveals that our audio-to-text descriptions provide valuable implicit visual context (e.g., "roaring engines and screeching tires" in car racing clips, "crowd cheering" in sports scenes), thereby enhancing discrimination among visually similar videos. This result confirms that the multimodal representations learned on OmniCVR transfer effectively to standard retrieval tasks.

Table 11: Zero-shot text-to-video retrieval performance on MSR-VTT. Our AudioVLM2Vec is trained only on OmniCVR and evaluated using its automatically generated audio-to-text descriptions as additional input. Best results are highlighted in **bold**.

| Model | R@1 | R@3 | R@5 | R@10 |
|---|---|---|---|---|
| VLM2Vec | 36.10 | 53.00 | 60.70 | 70.10 |
| AudioVLM2Vec (Ours) | **37.90** | **55.30** | **62.50** | **71.80** |
| Δ | **+1.80** | **+2.30** | **+1.80** | **+1.70** |

### C.4 DISCUSSION ON THE PLAUSIBILITY OF COMPOSITIONAL TRIPLETS

We clarify that our benchmark construction is strictly data-driven. We do not synthesize arbitrary instructions; instead, we first mine valid video pairs (Source, Target) from real-world distributions (e.g., HowTo100M, YouTube8M) based on semantic similarity. The modification text is generated post-hoc to describe the actual physical and acoustic differences observed between the clips. Thus, all "transformations" in OmniCVR reflect natural variations found in large-scale video corpora, ensuring that matching targets physically exist within the dataset.

### C.5 WHY DO "FULL-MODALITY" METHODS DIVERGE ON AUDIO-CENTRIC QUERIES?

Although ImageBind, OmniEmbed, and VLM2Vec all operate in multi-modal settings, their behaviors differ markedly once retrieval is driven *only* by audio changes.

**ImageBind.** ImageBind is capable of ingesting audio and text, yet in our audio-centric OmniCVR splits—where speech-guided queries constitute a majority-it struggles to capture the *lexical* and *semantic* content of speech. Its audio pathway is optimized for generic cross-modal alignment rather than fine-grained speech understanding; as a result, instructions that hinge on who spoke, what was said, or subtle speech-state changes (e.g., tone, intent) are often mapped to embeddings dominated by background acoustics or coarse timbral cues. This mismatch can even *mislead* retrieval under audio-centric composition, causing the search to prioritize scenes with similar ambient sounds while ignoring the intended speech-driven modification.

**OmniEmbed.** OmniEmbed can encode audio, but its token budget and fusion design weigh visual tokens much more heavily than audio tokens across the sequence. During joint pooling, the resultant embedding is therefore dominated by vision features, and audio contributes weakly to the final representation. Under audio-centric queries—where the visual stream is deliberately held constant—this imbalance suppresses precisely the information that differentiates the targets, yielding poor retrieval.

**VLM2Vec.** VLM2Vec does not include an explicit audio branch. Consequently, it performs competitively in vision-centric and integrated (vision+audio) settings—where the visual stream provides reliable discriminative cues and the composed instruction contains strong visual operators—but fails when the *only* changing factor is audio. In audio-centric queries, the absence of an audio encoder leaves the model with no path to ground the modification.

**AudioVLM2Vec (ours).** In contrast, our approach leverages **Qwen2-Audio** to first convert raw audio-especially speech-into fine-grained textual descriptions. These transcribed and summarized audio semantics are then injected into the VLM2Vec pipeline, where they are aligned with vision-language embeddings. This design bypasses the limitations of weak audio tokenization and ensures that speech content is represented in the same space as visual and textual cues. As reflected by the audio-centric results, AudioVLM2Vec achieves **77.2 R@1** and **94.2 R@10**, far beyond all other

full-modality methods, showing that preserving semantic detail from audio before fusion is a decisive advantage.

In summary, the failure modes align with architectural choices: (i) speech-heavy audio-centric composition penalizes models that lack *speech-aware* audio representations, (ii) token and pooling imbalance can *dilute* audio contributions, and (iii) omitting an audio branch altogether leads to systematic failure whenever sound is the only supervisory signal. By contrast, our design demonstrates that converting audio to rich textual descriptions before multimodal alignment allows audio to function as a true *first-class* compositional signal in OmniCVR.

## D  ADDITIONAL QUALITATIVE EXAMPLES

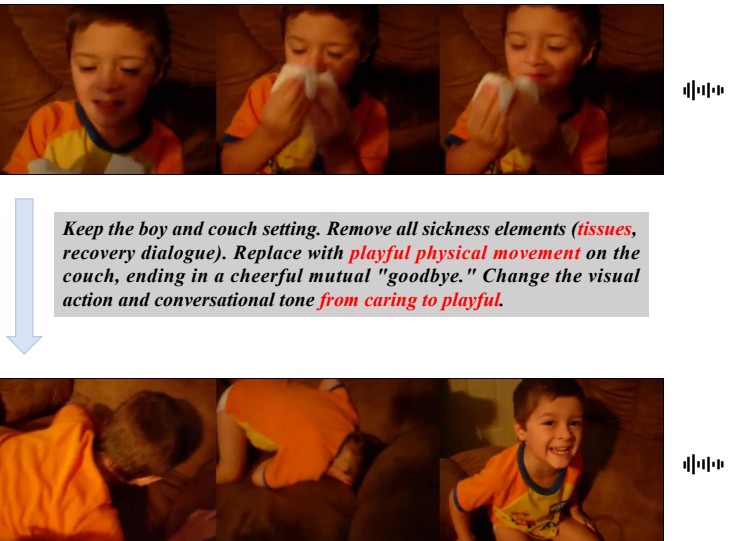

Figure 7: Qualitative example of modifying emotional tone and action. The query instructs the model to **remove sickness elements** (tissues, recovery dialogue) and **replace them with playful physical movement**, shifting the scene from caring to cheerful while maintaining the characters and setting.

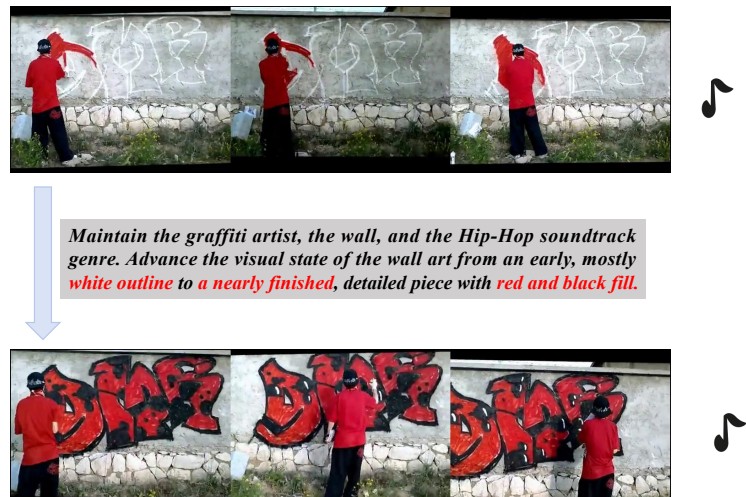

Figure 8: Qualitative example of temporal and state progression. The model is tasked to **advance the visual state** of the graffiti wall from an early outline to a *nearly finished, detailed piece*, while preserving the specific artist and Hip-Hop audio backdrop.

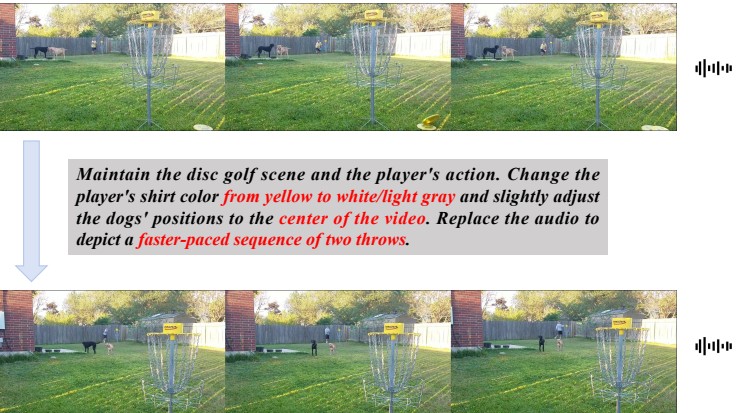

Figure 9: Qualitative example of fine-grained integrated modification. The instruction requires simultaneous changes in vision (**changing shirt color from yellow to white/light gray**) and audio (**depicting a faster-paced sequence of throws**), testing the model's ability to handle precise attribute.

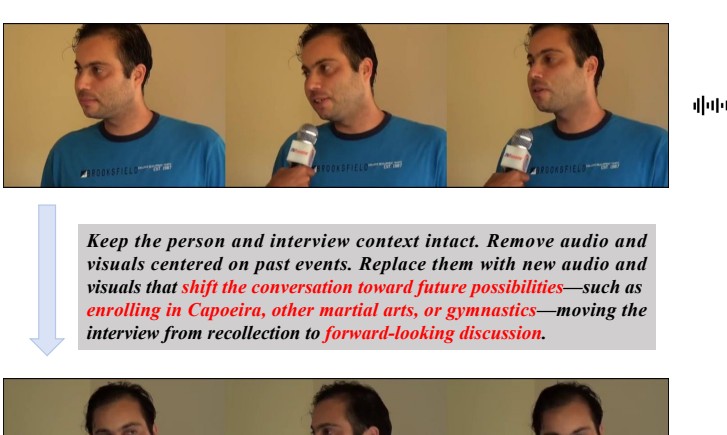

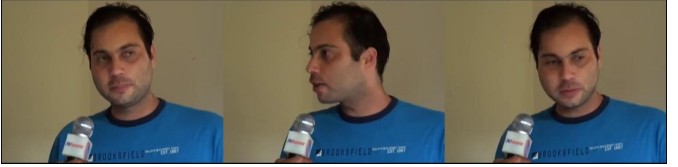

Figure 10: Qualitative example of semantic conversation shift. The query directs a transition from recollecting past events to discussing **future possibilities** (e.g., enrolling in Capoeira), requiring the model to understand the semantic content of the speech to retrieve the correct forward-looking segment.

We provide additional qualitative results to demonstrate the model's capability in handling diverse and complex compositional instructions. **Figure 7** illustrates a vision-centric transformation of emotional tone, shifting a scene from "caring recovery" to "playful interaction." **Figure 8** captures a temporal state progression, where the query directs the retrieval of a completed graffiti piece based on an early outline. **Figure 9** showcases a fine-grained integrated query, requiring simultaneous reasoning over visual attributes (shirt color change) and auditory events (pace of throws). Finally, **Figure 10** demonstrates a semantic conversation shift, where the model must distinguish between an interview segment about past recollections versus one focused on future aspirations.

## E    ETHICS STATEMENT

The OmniCVR benchmark proposed in this work is derived from existing public video datasets, including HowTo100M, MSR-VTT, VATEX, YouTube8M, YouCook2, and VALOR. We strictly adhere to the respective licenses and terms of use for these source datasets. The videos originate from publicly available content and do not involve private information belonging to individuals or organizations beyond what is already public.

For the human verification stage of our pipeline (Stage 3), we employed expert annotators to validate the quality of the modification instructions and video pairings. We ensured ethical working conditions and paid all annotators a fair hourly wage that exceeds the local minimum wage standards. Furthermore, during the generative annotation process involving Large Multimodal Models (LMMs), we applied safety filters to prevent the inclusion of harmful, offensive, or biased content in the generated text descriptions.

## F    REPRODUCIBILITY STATEMENT

We are committed to ensuring the reproducibility of our results and promoting further research in omni-modal retrieval.

- **Data Availability:** The OmniCVR benchmark, including the 160k+ curated clips, the 50k+ compositional triplets, and the gold-standard test set, will be made fully open-source to the public upon publication.

- **Code and Models:** Our proposed AudioVLM2Vec framework leverages open-source pre-trained weights (Qwen2-Audio and Qwen2-VL) and standard libraries. We will release the complete codebase, including data generation scripts, training code, and evaluation protocols.

- **Transparency:** To facilitate the reproduction of our dataset construction pipeline, we have provided the exact prompts used for annotation, triplet mining, and verification in Appendix G.

## G PROMPTS

Below are the prompts used for triplet mining.

---

**Prompt for generating chronological video captions using Qwen2.5-Omni**

You are a precise observer. Write one paragraph that describes ONLY what is directly visible and audible in the video, in strict chronological order with clear temporal markers.

For any notable action, break it into distinct stages and describe each in detail (e.g., starting slowly, changing technique, altering body position, adding props). If a stage is missing or unclear, state "not shown" or "unclear." This is the highest priority. Do not just state an activity; describe precisely how it unfolds and changes over time. Detail the sequence of movements. Example: Instead of "A person plays football," you must write: "A person begins by slowly dribbling a white and black ball with their right foot across a green field. They then transition to kicking it against a wall, and later, after lying on their back, they attempt to juggle the ball with their feet in the air."

**Rules:**

- No guesses or world knowledge.

- If something is uncertain, say "unclear".

- Include at least one sentence about **AUDIO** (speech, sound, or music). If silent, state "no audible speech; ambient silence/noise". If there were voices, specify who spoke, what was said in the video, and the emotion conveyed; if background music (BGM) is present, describe the genre and mood of the BGM; if there are other sounds, indicate their nature.

- Prefer concrete attributes (colors, materials, relative positions) over interpretations.

---

---

**Prompt for generating vision-only video modification instructions**

You are an expert in vision-only video understanding and creative language. Given two textual descriptions—[Source Description] and [Target Description]—generate a concise, natural-language instruction that tells someone how to modify the visual content of the "Source Video" to match the "Target Video".
Your instruction must:

- Focus only on **visible differences** (e.g., added/removed objects, different person, altered background, lighting, color tone, camera view, or scene layout).

- Completely ignore sound, audio, dialogue, music, or any non-visual information.

- Be phrased as a clear, direct command or user request.

- Start from the visual context of the Source Video.

---

**Now, generate a vision-only instruction for the following descriptions:**
[Source Description]: {source_desc}
[Target Description]: {target_desc}
**Generated Instruction:**

---

**Prompt for generating audio-only video modification instructions**

You are an expert in audio-only video understanding and creative language. Given two textual descriptions—[Source Description] and [Target Description]—generate a concise, natural-language instruction that tells someone how to modify the audio content of the "Source Video" to match the "Target Video".
Your instruction must:

- Focus only on **audible differences** (e.g., added/removed speech, changed speaker, background sounds, sound effects, music type, volume, or tone).

- Completely ignore any visual information (objects, actions, people, backgrounds, colors, lighting, or camera view).

- Be phrased as a clear, direct command or user request.

- Start from the audio context of the Source Video.

---

**Now, generate an audio-only instruction for the following descriptions:**
[Source Description]: {source_desc}
[Target Description]: {target_desc}
**Generated Instruction:**

---

**Prompt for generating video-to-video modification instructions**

You are an expert in creative language and video retrieval. Your task is to generate a concise, natural language instruction that describes how to modify a "Source Video" to become a "Target Video". You will be given two textual descriptions: [Source Description] and [Target Description].

Your instruction should:

- Identify the most salient difference(s) between the source and target (e.g., scene, action, object, person, setting).

- Be phrased as a clear command or user request.

- Be natural and easy to understand.

- Start from the context of the Source Video.

---

**Now, generate an instruction for the following descriptions:**
[Source Description]: {source_desc}
[Target Description]: {target_desc}
**Generated Instruction:**

---

**Prompt for generating detailed audio descriptions**

You are an expert audio analyst. Listen carefully to the given audio and provide a comprehensive description.
**Instructions:**

- First, identify the **main category** of the audio. Choose exactly one from: `human_speech`, `music`, `environmental_sound`, `event_sound`, `other`.

- Then, provide a **detailed explanation** according to the detected category:

  - **Human speech:** Transcribe the speech verbatim. Then identify the speaker's emotion (choose from: neutral, happy, sad, angry, fearful, surprised, disgusted).
  - **Music:** Describe genre, instruments, vocals (if any), rhythm, and the overall mood.
  - **Environmental sound:** Describe the environment, natural elements, and the acoustic atmosphere.
  - **Event sound:** Describe the specific event or action represented by the sound, including temporal sequence if clear.
  - **Other:** Provide as precise a description as possible of what is heard.

- Finally, summarize the entire audio in **one concise sentence**.

**Rules:**

- Do not use world knowledge beyond the audio itself.

- If something is uncertain or unintelligible, state "unclear".

- Prefer concrete acoustic attributes (e.g., pitch, tempo, loudness, timbre, clarity, background noise) over interpretations.

---

