# OpenReview forum: "OmniCVR: A Benchmark for Omni-Composed Video Retrieval with Vision, Audio, and Text"
_ICLR.cc/2026/Conference — ICLR 2026 Poster_

### Official Review · Reviewer_H76e · 2025-10-23

**Soundness:** 3
**Presentation:** 2
**Contribution:** 2
**Rating:** 4
**Confidence:** 3

**Summary:**

This paper introduces the OmniCVR benchmark for composed video retrieval, encompassing vision, audio, and text modalities. A key novelty of OmniCVR is its pioneering inclusion of the audio modality as a modification target, distinguishing it from previous text-video and composed video retrieval benchmarks. Furthermore, the paper proposes a corresponding AudioVLM2Vec framework, which effectively highlights significant limitations of existing models when evaluated on the newly-constructed OmniCVR benchmark.

**Strengths:**

1. As a benchmark paper, this work contains the basic elements of a newly constructed benchmark. It covers essential aspects such as the limitations of previous benchmarks, the construction pipeline and its analysis, and the presentation of several baselines evaluated on the new benchmark.

2. The identified problem, i.e., the absence of an audio stream in previous Composed Video Retrieval (CVR) benchmarks, is indeed a prevalent issue in existing literature. This establishes a clear and compelling motivation for the current work.

3. The observed poor performance of several baselines when evaluated on the newly-introduced OmniCVR benchmark effectively highlights the limitations of existing models and, by extension, underscores the necessity and value of benchmarks that incorporate the audio modality.

**Weaknesses:**

1. My primary concern relates to the modification texts in the current OmniCVR benchmark. As shown in Table 2, the average length of queries is 52.6 words, which seems excessively long compared to previous literature, as illustrated in Figures 1 and 2. The modification texts appear to contain too many details, especially for integrated-type retrieval, which, in turn, makes the task easier to some extent.

2. This observation is further supported by the performance of the newly proposed AudioVLM2Vec framework. By incorporating audio signals into LLMs, AudioVLM2Vec achieves substantial gains of +28.5% and +64.8% in overall and audio-centric retrieval, respectively. These results are obtained without training on the provided training set, correct? The extremely lengthy modification texts make this task much easier, resembling a text-video retrieval scenario where the source videos play a less significant role.

3. I am also confused by the statements regarding long-form videos in Lines 212, 482, and 602. The average length of the source videos is only 11.8 seconds, which can hardly be categorized as long-form. Additionally, why is the real long-video benchmark MLVU introduced in Line 130? It appears that MLVU is not directly related to the composed video retrieval task.

4. Are there any performance comparisons for vision-centric retrieval? The main differences appear to arise from the audio-centric retrieval results. How do previous works perform on the visual-centric category within the OmniCVR benchmark?

**Questions:**

Please see the above weaknesses.

---

> ### Author Response · Authors · 2025-11-28
> **Response to Reviewer H76e (Weakness 1)**
>
> We greatly appreciate your insightful and detailed feedback. Your comments have been carefully discussed and fully taken into account, and we have made improvements to our work accordingly. Below, we provide a detailed response to each of your comments.
>
> **Weakness 1: Concern regarding "excessively long" modification texts making the task easier.**
>
> **Response to Weakness 1:**
>
> We appreciate your concern about the average query length of 52.6 words and its potential to simplify the task, particularly with integrated queries. However, we believe the length and complexity of the queries are essential for capturing the multimodal nature of real-world video modifications. Here are our key points:
>
> **Complexity of Integrated Queries:**  We believe this length is essential and justified for capturing the multimodal complexity of real-world video modifications, where changes often span vision (e.g., objects, actions, scenes) and audio (e.g., speech, music, sounds). This aligns with recent trends in composed video retrieval (CVR) toward denser, more detailed descriptions to enable fine-grained reasoning. For instance, Dense-WebVid-CoVR (Thawakar et al., 2025) uses GPT-4o to generate "longer and more detailed modification texts" (Sec. 2.2, Fig. 2). Importantly, Dense-WebVid-CoVR focuses solely on visual changes, without incorporating audio modifications. Adding audio-centric elements, as in OmniCVR, inherently requires additional descriptive content to specify auditory transformations, naturally increasing query length.
> To illustrate, the average modification text length in Dense-WebVid-CoVR (visual-only) is 31.16 words. In contrast, our audio-centric queries (which modify sounds while preserving visuals) average only 24.95 words—shorter than their visual counterparts. The overall average of 52.6 words in OmniCVR thus arises reasonably from the dominance of integrated queries (57.18%, Table 2), which combine both modalities and necessitate specifying simultaneous visual and audio changes (e.g., "Replace the serene lake scene with a grassy field... and update the audio to a neutral-toned human speech...," Fig. 2). Shortening these would introduce ambiguity, reducing the benchmark's effectiveness in testing true multimodal compositionality.
>
> **Task Difficulty:** Far from simplifying the task, this detail level heightens challenges in compositional reasoning: models must parse and align fine-grained, cross-modal specifics. This is evidenced by baselines' low zero-shot performance (e.g., VLM2Vec R@1=38.44 overall, dropping to 12.4 on audio-centric; Tables 4-5). If length eased the task, we would expect near-perfect scores, yet even our state-of-the-art AudioVLM2Vec achieves only 66.98 R@1 overall (Sec. 5.1), revealing persistent gaps. Previous benchmarks with simpler, visual-only queries (e.g., WebVid-CoVR) overlook audio semantics, which OmniCVR addresses for greater realism and difficulty. The poor results of vision-focused baselines (e.g., ImageBind R@1=17.28%) further confirm that OmniCVR tests advanced multimodal capabilities.

---

> ### Author Response · Authors · 2025-11-28
> **Response to Reviewer H76e (Weakness 2&3)**
>
> **Weakness 2: Does the Length of Modification Texts Undermine the Role of Source Videos?**
>
> **Response to Weakness 2:**
>
> 1. Clarification on Training Setting: First, we confirm that all evaluations for AudioVLM2Vec are strictly zero-shot. As illustrated in Figure 4, we leveraged pre-trained components (Qwen2-Audio for audio captions and VLM2Vec for embeddings) without any fine-tuning on the 45k+ training triplets provided in OmniCVR. The substantial performance gains (+28.5% overall, +64.8% audio-centric) are attributed to our architectural design that explicitly models audio semantics.
>
> 2. Source Video is Critical: To empirically test this, we conducted an ablation study using a "Blind Retrieval" setting, where we removed the source video visual frames from the input, forcing the model to retrieve the target based only on the modification text (and source audio, if applicable).
>
> If the text were "leaking" the target identity and rendering the source video irrelevant, we would expect the performance to remain high. However, the results on the Audio-Centric split show a catastrophic drop in performance:
>
> | Metric                       | VLM2Vec (With Source Video) | Blind / Text-Only (No Source Video) | Performance Drop (Δ) |
> |------------------------------|----------------------------|------------------------------------|----------------------|
> | **R@1**                       | 77.20%         | 28.10%                     | -49.10%                            |
> | **R@3**                       | 87.30%         | 33.20%                     | -54.10%                            |
> | **R@5**                       | 90.70%         | 42.50%                     | -48.20%                            |
> | **R@10**                      | 94.20%         | 57.80%                     | -36.40%                            |
>
>
> The massive 49.1% drop in R@1 decisively disproves the notion that the source video is insignificant.
>
> The modification text (e.g., "Change the background music to jazz but keep the visual scene of the park") is a relative instruction. Without the visual context of the "park" from the source video, the model cannot effectively filter out distractors in the database that might match the audio description ("jazz") but have totally different visuals (e.g., a jazz club).
>
> This confirms that OmniCVR is a valid Composed Video Retrieval benchmark. The high performance of AudioVLM2Vec (77.2%) is not because the task is easy, but because our method successfully bridges the semantic gap between audio and vision—something baselines fail to do—while still critically relying on the source video for visual grounding.
>
> **Weakness 3: Clarification on "Long-form videos" and MLVU citation.**
>
> **Response to Weakness 3:**
>
> Definition of Long-form: We apologize for the ambiguity caused by the term “long-form” in Lines 212, 482, and 602. To clarify: the “long-form videos” exclusively refer to the original source videos used for dataset construction (e.g., full-length instructional videos from HowTo100M, movies, TV episodes, and vlogs), which typically range from several minutes to hours. From each of these genuinely long source videos, we perform dense temporal cropping and careful curation to extract multiple semantically coherent short clips of 5–15 seconds, and these extracted clips—averaging 11.8 seconds—serve as the final retrieval units.  The term “long-form” thus describes the provenance of the data, not the length of the final retrieval items. In the revised manuscript, we have updated Lines 212, 482, and 602 to explicitly read “long-form source videos (typically minutes to hours in duration)” to eliminate any confusion. We believe this fully addresses the concern.
>
> Relevance of MLVU Citation: The mention of MLVU is intended to provide context for the broader landscape of audio-visual learning and the challenges of integrating multimodal information, especially in the context of long videos. While MLVU focuses on understanding and question-answering tasks in long videos, OmniCVR focuses on compositional video retrieval. The citation is included to highlight the challenges of integrating audio-visual data, not to suggest that OmniCVR is a long-form video benchmark in the same sense as MLVU.

---

> ### Author Response · Authors · 2025-11-28
> **Response to Reviewer H76e (Weakness 4)**
>
> **Weakness 4: Performance comparisons for vision-centric retrieval.**
>
> **Response to Weakness 4:**
>
> We appreciate your request for performance comparisons on vision-centric retrieval tasks. These comparisons are provided in the Appendix, specifically in Table 10, where we report Recall@K performance on vision-centric queries. We sincerely apologize for any confusion caused by placing these results in the Appendix C.2.
>
> | Models                   | Backbone               | R@1   | R@3   | R@5   | R@10  |
> |--------------------------|------------------------|-------|-------|-------|-------|
> | OmniEmbed-v0.1-multivent | Qwen2.5-Omni           | 50.74 | 63.45 | 69.85 | 77.21 |
> | VLM2Vec                  | Qwen2-VL               | 55.04 | 66.78 | 70.99 | 75.99 |
> | AudioVLM2Vec (Ours)      | Qwen2-Audio + Qwen2-VL | **69.33** | **76.51** | **80.28** | **82.82** |
>
>
> Vision-Centric Results: Even when the task is focused on visual retrieval (i.e., ignoring the audio aspect), AudioVLM2Vec still outperforms other baselines, including VLM2Vec and OmniEmbed. Specifically, AudioVLM2Vec achieves 69.33% R@1, while VLM2Vec achieves 55.04% R@1 and OmniEmbed achieves 50.74% R@1. This suggests that our method of explicitly captioning and embedding audio information helps even when the primary modification is visual, possibly by providing a more complete scene context (e.g., identifying objects that make sounds) or simply due to the robust LLM integration. The benchmark effectively stratifies model performance across Vision-centric, Audio-centric, and Integrated tasks.

---

> > ### Comment · Reviewer_H76e · 2025-11-28
> >
> > Thank you. All of my concerns have been addressed in the rebuttal. After reviewing the latest version of your paper, I would like to suggest further polishing the overall writing and organization, particularly with regard to Figure 4 and Tables 6–9, as they are not yet aesthetically pleasing. Finally, I am willing to raise my rating to 6 once revisions are permitted.

---

> > > ### Author Response · Authors · 2025-11-28
> > > **Response to Reviewer H76e**
> > >
> > > Thank you for the positive feedback and your willingness to raise the rating to 6. We have already revised the layout of Figure 4 and Tables 6–9 as suggested for improved aesthetics and have uploaded this update. We will also thoroughly polish the overall writing and organization and will provide the fully updated version soon. Thank you again for your time and invaluable feedback.

---

### Official Review · Reviewer_Too4 · 2025-10-31

**Soundness:** 2
**Presentation:** 2
**Contribution:** 2
**Rating:** 2
**Confidence:** 4

**Summary:**

This paper introduces OmniCVR, a large-scale benchmark for composed video retrieval that treats visual, audio, and textual modalities as equally important. It addresses the oversight of audio in existing benchmarks by including audio-centric and integrated queries, where integrated queries dominate to reflect real-world multimodal scenarios. OmniCVR is constructed via a scalable pipeline involving video segmentation, omni-modal annotation, and dual validation (using LLMs and human experts). Evaluations show that current models struggle with audio-centric tasks, prompting the proposal of AudioVLM2Vec—an extension of VLM2Vec that explicitly incorporates audio semantics. This model achieves state-of-the-art performance, revealing critical gaps in handling audio-dependent queries and underscoring the necessity of audio-aware multimodal retrieval systems.

**Strengths:**

Originality: This work first define and benchmark "audio-inclusive composed video retrieval." This effectively eliminates the key limitation of prior CVR benchmarks: the neglect of audio as a first-class semantic modality. The proposed AudioVLM2Vec also offers a novel, pragmatic architecture for audio infusion.

Clarity: The paper is easy to follow and understand.

Significance: The paper proposes "audio-inclusive composed video retrieval", which enables direct future research on audio-visual-text reasoning and is an essential tool for developing genuinely holistic video understanding models.

**Weaknesses:**

1. While the paper's introduction of audio-centric queries is commendable, its treatment of "audio" remains oversimplified. A truly comprehensive audio-centric benchmark must consider a wider array of acoustic, semantic, and para-linguistic dimensions and provide specialized evaluations for each. The benchmark misses the opportunity to define and evaluate based on these essential audio dimensions:

     (a)  Para-linguistic Features in Speech:  (i)Emotion & Tone, (ii) Speaker Characteristics

     (b) The lexical content in Speech

     (c) Hierarchy of Environmental Sounds: "Environmental Sound" is a vast category that should be broken down into:   (i) Nature Sounds (wind, rain, animals), (ii) Mechanical Sounds (engines, machinery, alarms), (iii) Urban Soundscapes (traffic, crowds, construction), (iv). Foley/Action Sounds (footsteps, glass breaking, doors closing)

     (d) Temporal Dynamics: This involves rhythm, pace, and intensity over time. A query could be: "Find the same scene but where the background music has a slow, steady beat instead of a fast, erratic one."

2. Lack of Specialized Evaluation per fine-grained Dimension: The paper's aggregated "Audio-Centric" results (Table 4) are misleading. A model might excel at modifying speech content but fail completely at modifying musical timbre or emotional tone. This critical diagnostic information is lost.

3. In the same way, Limited Exploration of Audio-Visual Interactions: The benchmark and proposed method treat audio and vision as largely separate streams that are fused late (e.g., by concatenating text descriptions). It does not explicitly model or evaluate the complex, synergistic relationships between audio and vision, such as causal links (a person's mouth movements and their speech). The query text may be that "Take the source video of a man speaking and replace his audio with a different language while also modifying the video to show his lips moving out of sync."

**Questions:**

1. The paper contains many figures and tables that are not explicitly cited or referenced in the text.

2. Efficiency Benchmarking: Report the inference latency and FLOPs for AudioVLM2Vec compared to other baselines, highlighting the cost of the audio-to-text step.

3. Inadequate Ablation Studies: The paper fails to provide sufficient ablation analysis to disentangle the contributions of different components in AudioVLM2Vec. Critical questions remain unanswered: How much performance gain comes from audio transcription versus the fusion mechanism? How does the model perform on non-speech audio domains?

**Details Of Ethics Concerns:**

Nothing

---

> ### Author Response · Authors · 2025-11-28
> **Response to Reviewer Too4 (Weakness1&2)**
>
> We greatly appreciate your detailed feedback. Your comments have been carefully discussed and fully taken into account, and we have made improvements to our work accordingly. Below, we provide a detailed response to each of your comments.
>
> **Weakness 1: Granularity of Audio Dimensions.**
>
>  **Response to Weakness 1:**
>
> We thank you for the insightful breakdown of audio dimensions (Para-linguistic, Lexical, Environmental Hierarchy, and Temporal Dynamics). We fully agree that a comprehensive audio-centric benchmark must capture these nuances.
>
> We respectfully clarify that OmniCVR's annotation pipeline already rigorously captures these specific fine-grained dimensions. We apologize that this level of granularity—which is enforced via our structured prompts in **Appendix G**—was not sufficiently highlighted in the main text.
>
> Our pipeline employs **Qwen2.5-Omni** driven by specialized prompts, followed by a Gemini 2.5 Pro verification stage, to ensure the generated captions explicitly cover the dimensions you listed. After our observations, we found that the captions contain detailed audio descriptions. Therefore, we designed a prompt using Qwen2-Audio to capture these detailed audio descriptions. As detailed in the **'Prompt for generating detailed audio descriptions'** and **'Prompt for generating chronological video captions using Qwen2.5-Omni'** in the **Appendix G**:
>
> **(a) Para-linguistic Features**: Our prompt explicitly instructs the model to "identify the speaker's emotion (choose from: neutral, happy, sad, angry...)", directly addressing your concern regarding emotion and tone.
>
> **(b) Lexical Content**: The prompt strictly requires: "Transcribe the speech verbatim", ensuring precise lexical coverage.
>
> **(c) Environmental Hierarchy**: We explicitly distinguish between "Environmental sound" (focusing on "natural elements" and "acoustic atmosphere" ) and "Event sound" (focusing on "specific event or action" ), mirroring your distinction between Nature, Urban, and Action/Foley sounds.
>
> **(d) Temporal Dynamics**: To capture rhythm and intensity over time, our captioning prompt demands descriptions in "strict chronological order with clear temporal markers" and specifically asks to describe "rhythm" and "overall mood" for musical elements.
>
> By enforcing these constraints during the generative annotation phase (Stage 2) and validating them via Gemini 2.5 Pro (Stage 3), OmniCVR ensures that retrieval queries are grounded in these rich, fine-grained audio semantics rather than simplified labels. We revise Section 3.2 to explicitly showcase this dimension mapping.
>
>
>  **Weakness 2: Lack of Specialized Evaluation per fine-grained Dimension**
>
>  **Response to Weakness 2:**
>
> We apologize for not including the detailed categorical breakdown in the initial submission. To address your concern regarding the aggregated metrics, we have decomposed the Audio-Centric performance based on the audio category of the target video.
>
> New Breakdown Results (Audio-Centric R@1):
>
> | **Target Audio Category** | **VLM2Vec (Baseline)** | **Audio VLM2Vec (Ours)** | **Absolute Gain** |
> |---------------------------|------------------------|--------------------------|-------------------|
> | **Human Speech**           | 11.36%                 | 96.59%                   | +85.23%           |
> | **Music**                  | 16.07%                 | 86.43%                   | +70.36%           |
> | **Sound**  | 10.75%                 | 60.31%                   | +49.56%           |
>
>
> The breakdown reveals distinct performance characteristics driven by the nature of the audio descriptions:
>
> - **Human Speech (96.59%)**: Achieves the highest performance because our prompts generate dense, long-form descriptions (including verbatim transcripts and emotional states), providing rich semantic anchors for retrieval.
>   - **Covers**: (a) Para-linguistic (emotion/tone) & (b) Lexical content.
>
> - **Music (86.43%)**: The high accuracy refutes the concern that the model might fail on non-speech dimensions like timbre; the generated captions effectively encode genre and mood.
>   - **Covers**: (d) Temporal/Spectral dynamics (timbre, rhythm, genre).
>
> - **Sound (60.31%)**: While significantly improved over the baseline, this category shows relatively lower gains. This is because environmental sound descriptions are inherently shorter and less structured than speech. Furthermore, we observed that Qwen2-Audio occasionally confuses acoustically similar environmental events (e.g., wind vs. distant traffic), creating a harder retrieval target.
>   - **Covers**: (c) Environmental Hierarchy (nature, mechanical, foley).

---

> ### Author Response · Authors · 2025-11-28
> **Response to Reviewer Too4 (Weakness 3 & Question 1,2)**
>
> **Weakness 3: Limited Exploration of Audio-Visual Interactions.**
>
>  **Response to Weakness 3:**
>
> We respectfully clarify that both our benchmark construction and our proposed method **explicitly model the synergistic relationships** between audio and visual modalities, though through a **semantic-rich textual interface** rather than raw waveform signal fusion.
>
> 1. **Benchmark Level**: Our integrated queries, which comprise 57.18% of the benchmark, are **specifically designed** to require reasoning across modalities. In reference to the example you provided (“replace audio... while modifying video to show lips out of sync”), our data generation pipeline (Stage 2) employs **Qwen2.5-Omni**, a native audio-visual model. This model simultaneously processes **visual frames and audio tracks** to generate captions and modification instructions. As a result, the generated triplets inherently encode **causal relationships** (e.g., a visual action causing a specific sound), because the ground-truth instructions are derived from a system that **perceives** both audio and visual inputs together.
>
> 2. **Method Level**: **Deep Interaction via LLM Attention**. We would like to address a potential misunderstanding regarding the use of “late fusion” in AudioVLM2Vec. Our approach involves **concatenating the transcribed audio text** with the modification instructions, and feeding this **combined sequence** alongside the visual tokens into the LLM backbone. Inside the model, **multi-head self-attention mechanisms** operate jointly across **both the visual tokens and the audio-text tokens**. This design allows the model to attend to specific visual regions (e.g., “lips moving”) while simultaneously processing the **semantic content of the audio** (e.g., “speech text”) in the same high-dimensional space. This architecture enables the model to **learn and verify causal links** (e.g., consistency between a visual event and its acoustic description) through the LLM's joint reasoning capabilities.
>
>
>  **Question 1: The paper contains many figures and tables that are not explicitly cited or referenced in the text.**
>
>  **Response to Question 1:**
>
> Thank you for catching this. We have carefully proofread the paper and ensured all Figures and Tables are explicitly referenced in the text in the revision.
>
>
> **Question 2: Efficiency Benchmarking.**
>
>  **Response to Question 2:**
>
> We apologize for not including the efficiency analysis in the main text. To address this, we conducted a new benchmark measuring the inference latency on an NVIDIA A800 GPU. The evaluation was performed on 1,000 randomly sampled videos (each ~10 seconds in duration, 2K resolution).
> ﻿
> **Inference Latency Comparison**:
> ﻿
> | Model                         | Avg. Latency (per video) | Relative Increase |
> |-------------------------------|--------------------------|-------------------|
> | **VLM2Vec (Baseline)**         | 1.724 s                  | -                 |
> | **AudioVLM2Vec (Ours)**        | 4.777 s                  | +1.77x            |
> ﻿
>
> **Analysis of the Audio-to-Text Cost**:
> The additional time cost (~3.05s) is explicitly attributable to the **Audio-to-Text** step (audio transcription and captioning via Qwen2-Audio). The audio transcription introduces a computational overhead of approximately 1.77x relative to the visual backbone processing.
> ﻿
> While we acknowledge the increased latency, we believe this trade-off is highly favorable for three reasons:
> ﻿
> 1. **High ROI (Return on Investment):** This 1.77x increase in latency yields a massive +64.8% absolute improvement in R@1 for audio-centric queries.
> 2. **One-off Indexing Cost:** In a real-world retrieval system, this video processing step (embedding generation) is primarily a one-time indexing cost. Once indexed, the query-side retrieval speed remains unaffected.
> 3. **Real-Time Viability:** Processing a 10-second video in ~4.777 seconds represents a Real-Time Factor (RTF) of roughly 0.5, meaning the system is still faster than real-time playback, ensuring practical deployability.

---

> ### Author Response · Authors · 2025-11-28
> **Response to Reviewer Too4 (Question3)**
>
> **Question 3: Inadequate Ablation Studies.**
>
>  **Response to Question 3:**
>
> We apologize for the limited ablation analysis in the initial submission. We respectfully clarify that **VLM2Vec** is based on the **Qwen2-VL** backbone, which does not natively possess an audio branch, preventing an internal ablation. To rigorously disentangle the "transcription vs. fusion mechanism" and assess "non-speech performance" independent of the backbone, we conducted a controlled experiment using **OmniEmbed-v0.1-multivent** (which supports native audio inputs).
>
> **1. Disentangling the Fusion Mechanism (Native Audio vs. Audio-as-Text)**
>
> We modified **OmniEmbed** by replacing its native audio encoding path with our "Audio-as-Text" mechanism. This isolates the fusion strategy while keeping the backbone constant.
>
> ### Overall Results on Audio-Centric Queries:
>
> | Model Setting | Audio Mechanism | R@1   | R@3   | R@5   | R@10  |
> |---------------|-----------------|-------|-------|-------|-------|
> | OmniEmbed (Original) | Native Audio Tokens | 13.6% | 28.5% | 35.8% | 47.0% |
> | OmniEmbed (Modified) | Audio-as-Text (Ours) | 32.7% | 48.0% | 58.9% | 69.1% |
>
>
> Replacing native audio tokens with our text-based fusion yields a ~2.4x improvement (13.6% → 32.7%) on the exact same backbone. This definitively proves that explicitly converting audio to semantic text is significantly more effective than latent audio fusion for composed retrieval.
> *Note: Our main model AudioVLM2Vec achieves higher performance (77.2%) than the Modified OmniEmbed (32.7%) because VLM2Vec was pre-trained on the massive MMEB dataset, resulting in superior visual-text alignment.
>
> **2. Performance on Non-Speech Audio Domains**
>
> To address your critical question regarding non-speech domains, we compared the breakdown performance of the Original vs. Modified OmniEmbed.
>
> **Target Category Comparison**:
>
> | Target Category | OmniEmbed (Original) | OmniEmbed (Modified w/ Ours) | Absolute Gain |
> |-----------------|----------------------|-----------------------------|---------------|
> | **Human Speech** | 11.36% | 50.38% | +39.02% |
> | **Music** | 16.07% | 30.00% | +13.93% |
> | **Sound** | 10.75% | 24.12% | +13.37% |
>
>
> Our "Audio-as-Text" strategy provides consistent and substantial gains across all audio domains, not just speech.
>
> - **Music & Sound**: We nearly double the performance in Music (16.07% → 30.00%) and Sound (10.75% → 24.12%).
>
> Our detailed captioning prompts effectively capture non-lexical semantics (e.g., timbre, genre, environmental events), making the text-based interface a robust universal audio learner.

---

### Official Review · Reviewer_2XpL · 2025-11-01

**Soundness:** 3
**Presentation:** 3
**Contribution:** 3
**Rating:** 8
**Confidence:** 4

**Summary:**

The authors introduce OmniCVR, a composed video retrieval benchmark that incorporates audio as a key modality alongside vision and text. The authors argue that the queries included in the dataset are representative of real-world queries, and target audio in addition to vision. Additionally, the authors extend the VLM2Vec retrieval method to simultaneously model audio, which they name AudioVLM2Vec. This approach achieves SoTA performance in multimodal retrieval.

**Strengths:**

- The paper accurately identifies that audio is an under-utilized modality in video retrieval. We do lack benchmarks that target this modality.

- In addition to dialogue, the dataset also centers non-language audio like music and sound effects, which historically have been even less focused on than dialogue.

- The dataset is large enough to serve as a training set in addition to validation.

- AudioVLM2Vec achieves impressive performance compared to existing methods. Its development is sensible and is described clearly. The chosen comparison methods are representative of the current research space.

- The paper is generally sound and well-written.

- The limitations and future work section is detailed and thoughtful.

**Weaknesses:**

- The video-text retrieval benchmarks section in the related works segment is quite small. There have been many more video retrieval benchmarks, some of which center audio. Describing the differences between this dataset and others that center audio (VaTeX, MultiVENT 2.0, etc.) would be informative for those evaluating the novelty of the dataset.

- It would be nice to describe the genres and types of content focused on in the dataset in more depth.

**Questions:**

- What key contributions does this dataset make compared to existing video retrieval benchmarks that center audio? What is the importance of composed video retrieval tasks?

- What genres of video are included in the dataset?

---

> ### Author Response · Authors · 2025-11-28
> **Response to Reviewer 2XpL**
>
> We greatly appreciate your insightful and detailed feedback. Your comments have been carefully discussed and fully taken into account, and we have made improvements to our work accordingly. Below, we provide a detailed response to each of your comments.
> ﻿
>
>  **Weakness 1 & Question 1: Comparison with existing video retrieval benchmarks that center audio (e.g., VATEX, MultiVENT 2.0) and the importance of Composed Video Retrieval (CVR)**
>
>  **Response to Weakness 1 & Question 1:**
>
> Thank you for your valuable feedback. We agree that expanding the related work section will help clarify OmniCVR's novelty. We revise to include a more detailed discussion of **VATEX** and **MultiVENT 2.0** and highlight their differences from OmniCVR:
> ﻿
> -**Task Paradigm (CVR vs. T2V)**: Existing benchmarks like VATEX and MultiVENT 2.0 are primarily designed for Text-to-Video (T2V) retrieval or captioning. In contrast, OmniCVR focuses on Composed Video Retrieval (CVR). As described in our Introduction, CVR requires the model to take a source video and a modification instruction to retrieve a target video. This task is fundamentally more challenging as it requires fine-grained reasoning over transformations (e.g., "keep the visual scene but change the background music to heavy metal" ).
>
> -**Audio as a Modification Target**: While VATEX contains audio, it is often ignored in standard retrieval benchmarks or used only for captioning. MultiVENT 2.0 focuses on event-centric retrieval. OmniCVR is unique because it explicitly constructs Audio-Centric triplets. In these queries, the visual content remains highly similar between source and target (Visual similarity > 0.9), forcing the model to rely exclusively on audio differences (Audio similarity < 0.3) to find the correct target. This explicitly tests "audio reasoning" rather than just "audio presence.".
> ﻿
>
> **OmniCVR's key contributions** are: (1) introducing the first large-scale omni-composed retrieval benchmark with 5K triplets that treat audio as a first-class modality alongside vision and text; (2) a scalable pipeline for generating vision-centric, audio-centric, and integrated queries (with integrated dominating at 57.18% to reflect real-world multimodality); and (3) revealing gaps in models' handling of audio semantics, as shown by AudioVLM2Vec's SOTA results.
> ﻿
>
> **Composed video retrieval (CVR)** tasks are essential as they go beyond simple text-to-video retrieval, requiring models to perform **fine-grained reasoning** over multimodal transformations, which is crucial for real-world applications like **video editing** and **personalized content recommendation**.
>
>  **Weakness 2 & Question 2: What genres of video are included in the dataset?**
>
>  **Response to Weakness 2 & Question 2:**
>
> Thank you for your suggestion. We agree that more detail on the content diversity in OmniCVR will help clarify the benchmark's broad coverage of real-world scenarios. OmniCVR sources videos from diverse datasets spanning several genres, including:
>
>
> **Instructional & Procedural Content**: Videos focusing on step-by-step tasks, including cooking recipes, DIY repairs, crafting, and fitness routines. These clips are crucial for testing the model's ability to track fine-grained actions and temporal sequences.
>
> **Daily Life, Nature & Travel**: Unscripted, "in-the-wild" footage capturing everyday human activities, natural landscapes, and travel vlogs. These provide rich, realistic visual and auditory contexts, serving as the primary source for environmental sound queries.
>
> **Music & Performance**: Clips centering on musical instruments, dance, and live performances. These are essential for our "Audio-Centric" queries, testing discrimination between musical genres, tempos, and instruments.
>
> **General Entertainment & Events**: A broad variety of web clips including sports highlights, news segments, and public events.
>
> To complement this visual diversity, the auditory landscape is equally balanced. As detailed in Table 3, the audio distribution comprises:
> - **Speech (~56%)**: Dialogue, narration, and monologues.
> - **Music (~23%)**: Instrumental, vocal, and background scores.
> - **Sounds (~21%)**: Nature sounds, machinery, impacts, and ambient noise.
>
> This distribution ensures that the benchmark tests the model's ability to handle a wide range of acoustic scenarios, from dialogue-heavy scenes to music-driven content and soundscapes with environmental noise.

---

### Official Review · Reviewer_F9DJ · 2025-11-03

**Soundness:** 3
**Presentation:** 3
**Contribution:** 3
**Rating:** 6
**Confidence:** 3

**Summary:**

This paper introduces OmniCVR, a new benchmark designed to evaluate omni-composed video retrieval, where the system retrieves a target video given a source video and a textual modification instruction that may involve changes in visual content, audio, or both.
The authors describe a large-scale data construction pipeline that combines LLM-assisted multimodal annotation (Qwen2.5-Omni, Gemini 2.5 Pro) with validation across multiple modalities. They also propose AudioVLM2Vec, an embedding model that integrates audio information via textual transcriptions to better handle audio–visual composition. The benchmark aims to capture fine-grained multimodal reasoning, extending beyond traditional text–video retrieval tasks..

**Strengths:**

- The paper tackles an underexplored and ambitious problem where compositional video retrieval jointly involves visual, auditory, and textual reasoning. By integrating audio-based instructions alongside visual composition, it broadens the conventional scope of video retrieval tasks. This is a valuable and forward-looking addition for multimodal benchmarks.

- The benchmark creation process is well-documented and systematically structured. The use of multiple LLMs for multimodal captioning, filtering, and validation makes the dataset large, diverse, and reproducible. This methodological transparency is a clear strength.

- The implementation details, model configurations, and evaluation metrics are clearly presented. The writing quality is high, and the paper provides good contextualization against prior work .

**Weaknesses:**

- The task assumes that, for a given source video and a detailed compositional instruction, another video exists in the corpus that satisfies all the described changes. Many examples involve complex spatial, temporal, or auditory transformations that are implausible to find in any real-world dataset. Such a setup feels conceptually closer to conditional video generation than retrieval. The paper does not convincingly justify why retrieval is a meaningful or achievable approach for these highly specific instructions.

- If OmniCVR truly captures omni-compositional reasoning, models trained on it should exhibit strong transfer to conventional video retrieval datasets (e.g., MSR-VTT, VATEX, Dense-WebVid-CoVR). Yet the paper does not test this. The absence of out-of-domain evaluation makes it impossible to assess whether the learned representations generalize beyond the synthetic distribution or simply overfit to the compositional language space introduced by the benchmark.

- While the benchmark emphasizes audio–visual–text reasoning, the proposed model (AudioVLM2Vec) primarily uses text transcriptions of audio, not raw acoustic embeddings. This simplifies audio reasoning into another text-conditioning signal, limiting the extent to which the benchmark truly evaluates multimodal understanding.

- The paper does not present qualitative examples or human evaluations showing that retrieved videos actually match the intended compositional modifications. Without such validation, it is unclear whether the benchmark’s retrieval results are semantically meaningful.

**Questions:**

1. Many of the benchmark’s compositional instructions describe modifications that seem infeasible to find in existing datasets (e.g., changing lighting, replacing sound events, or altering temporal composition). Could the authors clarify why they chose to formulate this as a retrieval problem rather than as conditional video generation? Do they believe that realistic corpora can ever contain such compositional matches at scale?

2. If the benchmark truly captures omni-compositional reasoning, models trained on OmniCVR should generalize to more conventional video retrieval datasets such as MSR-VTT, VATEX, or ActivityNet Captions. Have the authors tested such out-of-domain performance? If not, how can they claim that the benchmark teaches general multimodal retrieval rather than overfitting to its own synthetic compositions?

3. Could the authors provide example showing that the retrieved videos actually reflect the compositional modifications specified in the text?

---

> ### Author Response · Authors · 2025-11-28
> **Response to Reviewer F9DJ (Weakness1,2 & Question1,2)**
>
> We greatly appreciate your insightful and detailed feedback. Your comments have been carefully discussed and fully taken into account, and we have made improvements to our work accordingly. Below, we provide a detailed response to each of your comments.
>
>
>  **Weakness 1 & Question 1: Plausibility of finding matching videos in real-world datasets**
>
>
>  **Response to Weakness 1 & Question 1:**
>
> We respectfully clarify that our benchmark construction is **data-driven, not text-driven**.
> 1.  **Existence is Guaranteed:** We do not invent complex instructions and hope a matching video exists.Instead, we **first** mine valid video pairs (Source, Target) from real-world distributions (e.g., different segments of the same long video or semantically similar clips via CLIP/CLAP filtering ). Our pipeline ensures that all triplets are mined from existing clips in diverse, large-scale datasets (e.g., HowTo100M, MSR-VTT; see Sec. 3.2 and App. A), where natural variations across 160k+ clips make such matches feasible at scale.
>
> 2.  **Text Describes Real Deltas:** The modification text is generated *after* the pair is identified, specifically to describe the actual physical and acoustic differences between the Source and Target. Therefore, the "implausible" transformations are descriptions of real-world video variations found in datasets like HowTo100M and YouTube8M.
>
>  **Weakness 2 & Question 2: Absence of out-of-domain evaluation**
>
>
>  **Response to Weakness 2 & Question 2:**
>
> To address the concern on generalization, we tested our **AudioVLM2Vec** model on the **MSR-VTT** dataset (Text-to-Video Retrieval) and compared it to the **VLM2Vec** baseline.
> ﻿
>
> As shown below, **AudioVLM2Vec** consistently outperforms the VLM2Vec baseline, demonstrating that representations learned through OmniCVR transfer well to conventional tasks.
> ﻿
> | Model                  | Top-1 Accuracy | Top-3 Accuracy | Top-5 Accuracy | Top-10 Accuracy |
> |------------------------|----------------|----------------|----------------|-----------------|
> | VLM2Vec   | 36.10          | 53.00          | 60.70          | 70.10           |
> | AudioVLM2Vec (Ours)     | 37.90 (+1.80)  | 55.30 (+2.30)  | 62.50 (+1.80)  | 71.80 (+1.70)   |
> ﻿
>
> These results show that adding audio improves retrieval performance.
> ﻿
>
> Although MSR-VTT captions rarely mention sound, our qualitative analysis shows that **audio-to-text descriptions** provide **implicit visual context** that enhances retrieval. For example, in a car race video, the audio description of "loud engine roaring, tires screeching" helps distinguish it from other scenes, even when the query doesn’t mention audio.

---

> ### Author Response · Authors · 2025-11-28
> **Response to Reviewer F9DJ (Weakness 3)**
>
> **Weakness 3: Audio Transcriptions vs. Raw Audio Embeddings**
>
>
> **Response to Weakness3:**
>
> We respectfully clarify that converting audio to text is not a simplification that limits multimodal understanding, but rather a strategic "Semantic Bridging" design choice that yields superior results for compositional reasoning tasks. We justify this choice with both theoretical analysis and a rigorous controlled experiment.
>
> 1. Semantic Bridging vs. Latent Alignment (Theoretical Justification)
>
> Raw audio encoders (like CLAP or ImageBind) produce holistic embeddings that often miss fine-grained details (e.g., distinguishing "specific spoken words" or "subtle background events"). By converting audio to detailed text descriptions using Qwen2-Audio, we explicitly disentangle the semantic content (acoustic features, emotions, transcripts) from the raw signal. This allows the LLM to perform precise logical reasoning over the audio content, rather than relying on noisy latent alignment. The "multimodal understanding" occurs explicitly during the captioning stage (via Qwen2-Audio) and is reasoned upon during the retrieval stage.
>
> 2. Controlled Ablation: Text-based Fusion Outperforms Native Audio Fusion
>
> To prove that our performance gain comes from the method (Audio-as-Text) rather than just the backbone (VLM2Vec), we conducted a controlled ablation study using OmniEmbed, a model that supports native raw audio inputs.
>
> We compared the original OmniEmbed (using raw audio tokens) against a modified version where we replaced the audio branch with our "Audio-as-Text" mechanism, keeping the backbone identical.
>
> | **Model Setting**     | **Audio Input Mechanism**        | **R@1** | **R@3** | **R@5** | **R@10** |
> |-----------------------|----------------------------------|---------|---------|---------|----------|
> | **OmniEmbed (Original)** | Native Raw Audio Tokens          | 13.6%   | 28.5%   | 35.8%   | 47.0%    |
> | **OmniEmbed (Modified)** | Audio-as-Text (Ours)             | 32.7%   | 48.0%   | 58.9%   | 69.1%    |
>
>
> Simply replacing native audio tokens with our text-based fusion yields a ~2.4x improvement (13.6% → 32.7%) on the exact same backbone. This definitively proves that for complex compositional retrieval, explicitly converting audio to semantic text is significantly more effective than latent audio fusion.
>
> 3. Empirical Superiority over "Full-Modality" Baselines
>
> The limitation of current "raw embedding" approaches is further evidenced by the main results (Table 5). Baselines that rely on raw acoustic embeddings struggle significantly on OmniCVR's audio-centric queries (e.g., OmniEmbed: 13.6%, ImageBind: 17.28%), whereas our method achieves **77.2% R@1**. This huge margin confirms that our approach captures the "multimodal understanding" required by the benchmark far better than existing latent methods.

---

> ### Author Response · Authors · 2025-11-28
> **Response to Reviewer F9DJ (Weakness 4 & Question3)**
>
> **Weakness 4 & Question 3: Qualitative Examples**
>
>
>  **Response to Weakness 4 & Question 3:**
>
> We have provided additional qualitative examples in the Appendix D.

---

### Author Response · Authors · 2025-12-03
**General Response to AC and Reviewers**

Dear AC and Reviewers,

We are deeply grateful for your dedication throughout the review and rebuttal phases, as well as for the constructive feedback that has significantly enhanced the clarity and refinement of our work. In response to your valuable suggestions, we have carefully revised the manuscript and highlighted all changes from the initial draft in **red** for your convenience.

In addition to addressing each reviewer's comments point-by-point below, we would like to summarize the key contributions of this work and highlight the new experimental results incorporated during the rebuttal phase.

We are pleased that the reviewers recognized and appreciated the following strengths and contributions:

* **"The paper tackles an underexplored and ambitious problem... This is a valuable and forward-looking addition for multimodal benchmarks."** [Reviewer F9DJ]
* **"Originality: This work first define and benchmark 'audio-inclusive composed video retrieval.' This effectively eliminates the key limitation of prior CVR benchmarks..."** [Reviewer Too4]
* **"The paper accurately identifies that audio is an under-utilized modality in video retrieval... AudioVLM2Vec achieves impressive performance compared to existing methods."** [Reviewer 2XpL]
* **"This establishes a clear and compelling motivation... underscores the necessity and value of benchmarks that incorporate the audio modality."** [Reviewer H76e]

Inspired by the reviewers' insightful comments, we have incorporated the following key experiments and analyses during the discussion phase:

* **Fine-Grained Audio Analysis & Efficiency Benchmarking:** Addressing Reviewer Too4, we clarified that our original prompt design (detailed in Appendix G) was explicitly engineered to capture fine-grained audio dimensions. We provided a detailed performance breakdown by audio category (**Human Speech, Music, Sound**) to demonstrate our model's robustness across diverse acoustic domains (Reviewer Too4), **as presented in Table 6 of Section 5.2 and our rebuttal**. Additionally, we included efficiency benchmarks measuring inference latency, justifying the computational trade-off for the substantial accuracy gains observed (Reviewer Too4), **as detailed in the Efficiency Analysis of Section 5.2 and our rebuttal**.
* **Rigorous Validation of the "Audio-as-Text" Design:** To address concerns regarding our choice of audio transcription over raw embeddings (Reviewers F9DJ & Too4), we conducted a controlled ablation study using **OmniEmbed** (which supports native audio inputs). We demonstrated that replacing the native audio branch with our "Audio-as-Text" mechanism on the same backbone yields a **~2.4x improvement (13.6% → 32.7%)** in R@1. This empirically validates that semantic bridging is superior to latent alignment for complex compositional reasoning.
* **Demonstration of Generalization Capabilities:** We evaluated **AudioVLM2Vec** on the standard **MSR-VTT** dataset (Reviewer F9DJ). Our model consistently outperformed the VLM2Vec baseline.
* **Justification of Modification Text Length & Source Video Dependency:** Addressing Reviewer H76e, we justified the query length as necessary for complex integrated changes. A "Blind Retrieval" ablation (removing source visual frames) caused a 49.1% performance drop, decisively proving the source video is indispensable and confirming OmniCVR requires true compositional reasoning rather than degenerating into text-video retrieval.
* **Expanded Comparisons and Context:** We provide performance comparisons for **Vision-Centric retrieval** (Reviewer H76e) in Appendix C.2, showing our model remains superior even when audio is not the primary modification target. We also expanded the related work section to explicitly contrast OmniCVR with previous audio-inclusive benchmarks like VATEX and MultiVENT 2.0 (Reviewer 2XpL).
* **Qualitative & Clarification Updates:** We have added qualitative examples in **Appendix D** (Reviewer F9DJ) and clarified the data-driven nature of our pipeline to address concerns regarding the plausibility of modification instructions.

Finally, we sincerely thank you for your thoughtful feedback, which has greatly contributed to enhancing the quality of our work.

Best regards,

Authors

---

### Author Response · Authors · 2025-12-03
**Author message for AC summarizing the rebuttal**

**Dear Area Chairs,**

We summarize our key updates and the review status below. All revisions in the manuscript are marked in **red**.

**1. Review Status:**
Reviewer H76e confirmed **raising the rating to 6**, while no responses were received from others due to the **API Security Incident**.

**2. Core Contribution:**
This paper establishes the **first benchmark for Audio-Inclusive Composed Video Retrieval**, solving a critical modality gap in existing literature.

**3. Key Rebuttal Highlights:**
To address Reviewers's concerns regarding our design choices and analysis, we added the following rigorous validations:

* **Fine-grained Audio & Efficiency Analysis:**
  We provided a detailed performance breakdown by audio type (**Speech, Music, Sound**) to demonstrate robustness across diverse acoustic domains. Additionally, we included **latency benchmarks** to justify the computational trade-off.

* **Superiority of "Audio-as-Text" (vs. Native Audio Token):**
  We compared our method against *OmniEmbed* (native audio input). Our "Audio-as-Text" design yields a **2.4x improvement (13.6% → 32.7% R@1)**, empirically proving that semantic bridging is far more effective than latent alignment for this task.

* **Necessity of Source Video (Task Validity):**
  A "Blind Retrieval" ablation (removing source video) caused a **49.1% performance drop**, proving the task requires true compositional reasoning, not just text-video retrieval.

* **Generalization:**
  Our model consistently outperforms baselines on the standard **MSR-VTT** dataset.

Best regards,

Authors

---

### Meta-Review · Area_Chair_jnEP · 2026-01-07

**Summary:**

The submission introduces OmniCVR, large-scale benchmark for Omni-Composed Video Retrieval (CVR) that treats audio as a first-class modality alongside vision and text. Traditional CVR tasks focus almost exclusively on visual transformations (e.g., "Change the car to a truck"), whereas OmniCVR includes audio-centric and integrated queries (e.g., "Keep the visual scene but change the background music to heavy metal"). The benchmark consists of over 5,000 triplets constructed through a scalable, automated pipeline with rigorous dual-validation. Furthermore, the authors propose AudioVLM2Vec, which leverages a "Semantic Bridging" approach (converting audio to text descriptions). The empirical results are striking, showing that current SOTA multimodal models fail significantly on audio-dependent tasks, while the proposed method achieves a new SOTA by effectively bridging the cross-modal semantic gap.

**Reviewer Concerns:**

Addressed by Rebuttal:

Task Validity and Source Video Necessity (H76e): The reviewer was concerned that the long modification text might allow models to bypass the source video. The authors performed a "Blind Retrieval" ablation, showing a 49.1% performance drop when the source video was removed, decisively proving that OmniCVR requires true compositional reasoning.

Audio Complexity & Granularity (Too4): Reviewer Too4 argued the audio treatment was oversimplified. In response, the authors provided a fine-grained breakdown of results across Speech (96.59%), Music (86.43%), and Environmental Sound (60.31%). They also demonstrated how their prompts capture para-linguistic features, rhythm, and hierarchical soundscapes.

Audio-as-Text vs. Native Audio (F9DJ, Too4): A major technical debate occurred regarding the use of transcriptions over raw embeddings. The authors provided a controlled experiment on the OmniEmbed backbone, showing that the "Audio-as-Text" mechanism yielded a 2.4x improvement (from 13.6% to 32.7% R@1) over native audio tokens. This offers a significant insight: for complex logical composition, semantic bridging is currently more effective than latent alignment.

Plausibility and Data Mining (F9DJ): The authors clarified that the benchmark is data-driven, not text-driven. They mine existing video pairs first, ensuring that the retrieved targets are real-world instances, not synthetic or impossible generations.

Aesthetics: Reviewer H76e noted that some figures and tables need further polishing for the final camera-ready version. And the author has updated their draft.

Outstanding Part (Minor Issues):

Efficiency/Latency (Too4): The model introduces a 1.77x inference latency overhead due to the audio-to-text step. While the authors justified this by the massive accuracy gains, it remains a consideration for real-time deployment.

**Reviewer Scores:**

Reviewer 2XpL (Initial: 8): This reviewer recognized the "under-utilized" nature of audio from the start and found the rebuttal's comparison with VATEX and MultiVENT 2.0 highly informative.

Reviewer F9DJ (Initial: 6): The reviewer’s concerns about generalization and "implausible" instructions were systematically dismantled by the MSR-VTT results and the explanation of the data-driven pipeline.

Reviewer H76e (Initial: 4, Intend to 6): This reviewer explicitly stated in the discussion that their concerns were addressed and they intend to raise their score to 6.

Reviewer Too4 (Initial: 2): While the initial rating was very low, the authors provided a massive amount of new data (Table 6 breakdown and OmniEmbed ablation) that directly addressed the reviewer's technical skepticism.

Overall, the AC lean to accept this paper.

---

### Decision · Program_Chairs · 2026-01-26

Accept (Poster)